# Using Arctic ice mass balance buoys for evaluation of modelled ice energy fluxes

Alex West[1], Mat Collins[2], Ed Blockley[1]

[1]Met Office Hadley Centre, FitzRoy Road, Exeter EX1 3PB
[2]Centre for Engineering, Mathematics and Physical Sciences, University of Exeter, Stocker Rd, Exeter EX4 4PY

**Abstract.** A new method of sea ice model evaluation is demonstrated. Data from the network of Arctic Ice Mass Balance buoys (IMBs) is used to estimate distributions of vertical energy fluxes over sea ice in two densely sampled regions - the North Pole and Beaufort Sea. The resulting dataset captures seasonal variability in sea ice energy fluxes well, and captures spatial variability to a lesser extent. The dataset is used to evaluate a coupled climate model, HadGEM2-ES, in the two regions. The evaluation shows HadGEM2-ES to simulate too much top melting in summer and too much basal conduction in winter. These results are consistent with a previous study of sea ice state and surface radiation in this model, increasing confidence in the IMB-based evaluation. In addition, the IMB-based evaluation suggests an additional important cause for excessive winter ice growth in HadGEM2-ES, lack of sea ice heat capacity, which was not detectable in the earlier study.

Uncertainty in the IMB fluxes caused by imperfect knowledge of ice salinity, snow density, and other physical constants is quantified (as is inaccuracy due to imperfect sampling of ice thickness) and in most cases is found to be small relative to the model biases discussed. Hence the IMB-based evaluation is shown to be a valuable tool with which to analyse sea ice models, and by extension better understand the large spread in coupled model simulations of present-day ice state. Reducing this spread is a key task both in understanding the current rapid decline in Arctic sea ice, and in constraining projections of future Arctic sea ice change.

## 1. Introduction

Evaluation of sea ice simulations using metrics based on sea ice extent (e.g. Stroeve et al, 2012; Wang and Overland, 2012) is known to be an imperfect method of assessing models (Notz, 2015). This is partly because of the very high interannual variability of sea ice extent (Swart et al, 2015), but also because it does not address the accuracy of the many variables influencing sea ice extent, in which compensating errors may be present. Sea ice volume, evaluated for CMIP5 by Stroeve et al. (2014) and Shu et al. (2015), is less sensitive to internal variability (Olonscheck and Notz 2018) but is also driven by multiple complex processes and so is equally susceptible to compensating errors. These issues hinder understanding of the very large spread in modelled present-day sea ice simulation. In turn, this increases the uncertainty in future projections of Arctic sea ice, which has declined rapidly over the past 30 years both in extent and volume (Lindsay and Schweiger, 2015; Kwok, 2018). In this study, we present a new, complementary method of evaluating sea ice simulation, motivated by the following reasoning.

The proximate driver of sea ice volume is sea ice mass balance. In turn, sea ice mass balance is driven by energy
balance at the upper and lower surfaces of the snow-ice column. Energy balance is driven partly by external factors
in the atmosphere (radiative fluxes, upper air temperature) and ocean (ocean heat flux) but also by sea ice
thermodynamics (temperature, albedo and conduction). Finally, the sea ice thermodynamics, in turn, are partially
driven by the sea ice state itself (area and thickness), closing the two causal chains known as the thickness-growth
feedback and the surface albedo feedback (Figure 1). Ideally, for any model, the entire sea ice causal chain would
be evaluated, along with the external drivers, greatly increasing understanding of why a particular sea ice state is
modelled.
Large-scale evaluation of ice mass balance, ice thermodynamics, and energy balance at the lower surface of the
ice, has not to date been performed for any model. This is largely because these quantities cannot yet be measured
remotely but must instead be measured in-situ using systems of instruments frozen into the ice. In particular, a
device called an Ice Mass Balance buoy (IMB) measures mass balance at the upper and lower surfaces of the
snow-ice column, and temperature profiles within the ice, at simultaneous locations (Perovich and Richter-Menge,
2006). Data from individual IMBs has been used to estimate sea ice energy fluxes such as conduction and ocean
heat flux in the past (e.g. Perovich et al, 2002; Lei et al, 2014; Lei et al, 2018). However, IMB data has not yet
been used to directly evaluate a climate model, due to the large disparity in the relevant spatial and temporal
scales.
In this study, we use data from the whole IMB network to perform a large-scale evaluation of sea ice mass balance
and thermodynamics in a coupled climate model. Monthly mean fluxes of top melt, top conduction, basal
conduction and ocean heat flux are calculated from temperature and elevation data obtained from 104 IMBs
released between 1993 and 2015: the resulting observational dataset, and the code used in its production, are
published alongside this study, with references given in the code availability and data availability sections below.
This dataset is then used to evaluate the sea ice in a coupled climate model (HadGEM2-ES, part of the CMIP5
ensemble) in two densely sampled regions of the Arctic, the North Pole and the Beaufort Sea. Modelled and IMB-
measured fluxes are restricted to each region in turn: distributions of fluxes in each month are compared, and
likely model biases identified. The results of the IMB-based evaluation are compared with a previous evaluation
of the sea ice state and surface radiation in the same model (West et al., 2019): the results are found both to be
consistent with the results of this study, and to enhance understanding of the first evaluation.
The manuscript is structured as follows. In section 2, the IMB data, and the process by which vertical energy
fluxes are calculated from the data, are described. In section 3, the IMB flux dataset is described: seasonal, spatial
and interannual variability is discussed, and uncertainty in the IMB fluxes due to various parameters used in the
analysis is examined. In section 4, the data are used to evaluate HadGEM2-ES, and the results are interpreted in
the context of West et al. (2019). In section 5, the representativity of the IMB fluxes is discussed. In section 6,
conclusions are presented.
**2. Calculating monthly-mean energy fluxes from the IMBs**
The ice mass balance buoy (IMB; Perovich and Richter-Menge, 2006) is a system of instruments frozen into a sea
ice floe, allowing the simultaneous measurement of surface and base elevation, internal ice temperature (usually
at 10cm resolution), and position; many also measure surface air pressure and temperature. A diagram of an IMB
is shown in Figure 2. An IMB provides, by design, measurements of sea ice thickness, and of surface and basal
mass balance, via the measurements of surface and base elevation. Fluxes of conduction can also be estimated
from the ice temperature data (e.g. Perovich and Elder, 2002), although uncertainty is considerable due to lack of
knowledge of ice salinity. In particular, the thermodynamics and basal elevation measurements can be combined
to estimate ocean heat flux (Lei et al., 2014).
Data from the 104 IMBs deployed by the Cold Regions Research and Engineering Laboratory (CRREL) are stored
in a series of comma-delimited CSV files at http://imb-crrel-dartmouth.org/results/ (Perovich et al, 2020). The
buoys were deployed between 1993-2017; spatial coverage is mainly in the North Pole and Beaufort Sea regions
(Figure 3). The buoys are identified by the year of deployment followed by a letter, for example '2012L'. Buoy
lifetimes range from 4 days (2015C) to 20 months (2006C) with an interquartile range of 4-11 months. All buoys
report time series of ice base elevation, snow/ice surface elevation, latitude, longitude, as well as a collection of
ice temperature time series taken at a number of vertical positions above, within and below the ice. In general,
temperature profiles are reported at very high temporal resolution, hourly or bi-hourly, and tend to be noisy, with
much high-frequency variability. From 2006 onwards, elevation data are reported at similarly high resolution, but
before 2006 are reported much less frequently, with intervals of a week or more between measurements.
As most analysis of the data depends on the ability to perform arithmetic operations on different series, it was
necessary to produce data series at consistent points in time for each buoy. To this end, modified elevation data
series were produced at times coincident with the temperature measurements, using either interpolation (where
there were fewer than 3 measurements in the 2-day period centred on the time in question) or a binomially-
weighted mean (where there were 3 or more measurements in this period). This regularisation process is illustrated
in Figure 4. Although a more advanced optimal interpolation scheme would likely produce more accurate time
series, inspection of individual data series shows that the current scheme produces data that is sufficiently realistic
for the purposes of this study. For example, linear interpolation produces unrealistic sharp changes in the time
derivative of elevation, but the effect of these on monthly mean elevation change, the derived variable used in this
study, is likely to be very small.
The set of elevation measurements provided also varies between buoys, necessitating some processing before full
regular time series of surface elevation, snow thickness, interface elevation, ice thickness and base elevation can
be obtained. Some later buoys do not report surface elevation directly, but report snow-ice interface elevation and
snow depth, which must be summed to obtain the surface elevation. A more difficult problem is presented by the
earlier buoys, which tend to produce data of surface and base elevation only. Snow-ice interface elevation must
therefore be deduced from surface and base elevation, by a process illustrated in Figure 5. Iterating through the
times of observation $t_1,\ldots,t_n$, the interface elevation $z_{\mathrm{int}}(t_1) = 0m$ by construction, as the thermistor string is
always referenced to the snow-ice interface at the time of deployment. At time $t_i$, if $z_{\mathrm{int}}(t_{i-1}) \leq z_{sfc}(t_i)$, where
$z_{sfc}(t_i)$ represents surface elevation of the snow-ice column, we set $z_{\mathrm{int}}(t_i) = z_{\mathrm{int}}(t_{i-1})$; but if
$z_{\mathrm{int}}(t_{i-1}) > z_{sfc}(t_i)$ we set $z_{\mathrm{int}}(t_i) = z_{sfc}(t_i)$. In this way, the interface elevation changes only when top
melting of ice is detected, i.e. when the surface elevation is judged to fall below the interface elevation estimated
for the previous time of observation.
This method would fail in the presence of ice flooding and snow-ice formation (e.g. as documented by Provost et
al, 2017). However, while snow-ice formation is known to occur in some areas sampled by the IMBs (particularly
in the North Pole region, e.g. Rösel et al, 2018), it is almost certainly a rare event in the IMB dataset. This is
because the snow layer is almost always sufficiently thin relative to the ice layer that snow-ice formation is
unlikely from hydrostatic principles. There are four instances when snow depth becomes sufficiently large that
snow-ice formation is a possibility, but these are always associated with failure of other sensors, such that the
associated data does not reach the final dataset produced in this study.
Processing the temperature data is also necessary. Instances of air, ice or ocean temperature data that are obviously
wrong occur very frequently, usually characterised by sudden step changes in the temperature measurements at
single, or multiple layers, that are inconsistent with simultaneous measurements in other layers, often to physically
unrealistic values. The incorrect values can be caused by failure of the sensors or the datalogger, or by an inability
to communicate data to the receiving satellite (Donald K. Perovich, personal communication). In most cases,
wrong values occurred in large groups that were difficult to identify with automatic data processing, and therefore
had to be identified by inspection and removed. From the processed temperature and elevation data, monthly mean
fluxes of top melt, top conduction, basal conduction and ocean heat flux were produced in the following way.
Throughout this study, the sign convention is that a positive value denotes a downwards flux, and vice versa.
*Top melting of ice and/or snow.* This flux, commonly reported by models, represents the total energy gain by sea
ice (snow) in a grid cell over the course of a month associated with melting of ice (snow) at the upper surface. It
is estimated from the IMBs using the surface elevation series. A change between two adjacent daily data points
in surface elevation is judged due to top melting if and only if the change is negative, and the surface temperature
is above a threshold value (-2°C). The energy gain associated with the melting is calculated by multiplying the
elevation change by ice or snow density, depending on whether the snow depth is nonzero, and by specific latent
heat of fusion of ice (all parameters are defined below). The daily top melt estimates are then averaged to obtain
monthly mean top melt.
*Top conductive flux.* This flux is defined as the conduction from the snow/ice surface into the ice interior. In this
study it is calculated using temperatures in the top 50cm of the snow-ice column. Where this layer lies entirely
within snow (ice) the conductive flux is calculated as the temperature gradient across the layer, determined by a
linear fit, by snow (ice) conductivity: values of snow and ice conductivity used are defined below.
In many cases, however, the top 50cm is located partly within snow and partly within ice. Because snow
conductivity tends to be much lower than ice conductivity, the snow-ice interface is usually associated with a
sharp change in gradient that renders a linear fit meaningless. In these cases, the top conductive flux is determined
by a linear fit through the same layer, using an 'adjusted' temperature profile:
$$T_{adj}(z) = \begin{cases} \mu T(z) + (1-\mu)T_{int-r} & z > z_{int} \\ T(z) & z \le z_{int} \end{cases} \quad (1)$$
where $z_{int}$ is the elevation of the snow-ice interface, $T_{int-r}$ is temperature 5cm below the interface, and
$\mu = k_{ice}/k_{snow}$ where $k_{ice}$ and $k_{snow}$ are ice and snow conductivity respectively. Physically, $T_{adj}$ represents the
temperature profile that the snow-ice column would have, if the snow was converted to ice, $T_{int-r}$ remained the
same, and the vertical conductive fluxes remained the same. The effect of the adjustment is to 'straighten' the
profile by rotating the profile section located in the snow about $T_{int-r}$, by a factor determined by the ratio of
conductivities $\mu$. A linear fit is then taken through a layer 0-50cm below the snow surface, and multiplied by $k_{ice}$
to produce estimates of instantaneous top conductive flux. These are then averaged to obtain monthly means. The
process is illustrated in Figure 6.
*Basal conductive flux.* This flux is defined as the conduction from the ice base into the ice interior. As an important
component of the energy balance at the ice base it has frequently been estimated from individual buoys in ocean
heat flux calculations. Typically, temperature gradients at the ice base are small due to higher salinities here (e.g.
Schwarzacher, 1959), with correspondingly higher heat capacities and lower conductivities; hence previous
studies have commonly used a reference layer of a fixed thickness above which the basal conduction is estimated.
In this study we use the approach of Lei et al. (2014), and calculate the basal conduction by taking temperature
gradients across a layer 40cm-70cm above the ice base, illustrated in Figure 2. In section 3.3 we examine the
sensitivity of the derived fluxes to changes in the elevation of this reference layer, amongst other parameters. As
above, the instantaneous values were averaged to a monthly mean.
*Ocean heat flux.* This flux is defined as the diffusive heat flux arriving at the ice base from the ocean beneath. In
theory, it can be calculated as the residual of the basal conductive flux and the latent heat of melting/freezing at
the ice base. However, using the basal conductive flux as defined above it is necessary also to take into account
the sensible heat uptake of the intervening layer (the 'buffer zone'), 0-40cm above the ice base, illustrated in
Figure 2. The ocean heat flux can then be written as
$$F_{ocn} = F_{condbot} - F_{sens} - F_{lat} \quad (2)$$
as in Lei et al (2014).
The basal conductive flux $F_{condbot}$ is defined as above. Monthly mean $F_{sens}$, the sensible heat flux in the 0-40cm
layer, is calculated as the average of daily heat uptake rates obtained by taking linear fits through all temperature
points within 1 day of a given time instant for all vertical points in this layer, summing these (weighted according
to layer thickness), and multiplying by ice density and heat capacity, defined below. Finally, monthly mean latent
heat of melting at the ice base, $F_{lat}$, is calculated from the base elevation time series, by multiplying daily
differences in elevation by specific latent heat of fusion.
The calculation of thermodynamic parameters is now described. In this study, we take the approach of using a
'standard' set of thermodynamic parameters to calculate the main dataset of energy fluxes, demonstrated in
sections 3.1 and 3.2 below, and subsequently evaluate sensitivity to the values of these parameters in section 3.3.
Ice density $\rho_{ice}$, snow density $\rho_{snow}$ and latent heat of melting $q_{fus}$ are set to 917 kgm$^{-2}$, 330 kgm$^{-2}$ and 3.34 x
105 Jkg$^{-1}$ respectively, the standard values used by the sea ice model CICE (Hunke et al, 2013).
Ice conductivity is defined after Maykut and Untersteiner (1971) as
$$k_{ice} = k_{fresh} + \frac{\beta S}{T} \qquad (3)$$
where $S$ and $T$ are ice salinity and temperature respectively, $k_{fresh} = 2.03 Wm^{-1}K^{-1}$, the conductivity of fresh
ice, and $\beta = 0.13 Wm^{-1}$ is an empirically determined constant representing the effect of brine pockets on
conductivity. For the calculation of the top conductive flux, a practical salinity of 1.0 is used, while the temperature
used is that of the snow-ice interface. For the calculation of the basal conductive flux, a practical salinity of 4.0 is
used, multiplied by the mean value of $1/T$, where the average is taken over the time period in question and the
layer 40-70cm above the ice base.
Specific heat capacity is defined after Ono (1967) as

$$c_{ice} = c_{fresh} + \frac{q_{fresh}\mu S}{T^2} \qquad (4)$$

where $c_{fresh} = 2106 Jkg^{-1}K^{-1}$ is the specific heat capacity of fresh ice, $q_{fresh} = 3.34 \times 10^5 Jkg^{-1}$ the
specific latent heat of fusion of fresh ice, and $\mu = 0.054K$ the ratio between water salinity and freezing
temperature. In calculating sensible heat uptake at the ice base, again a practical salinity of 4.0 is used, multiplied
by the mean value of $1/T^2$, where the average is taken over the time period in question and the layer 0-40cm
above the ice base.
Ice salinity must also be taken into account when calculating latent heat of freezing and melting. The energy
required to melt a given volume of sea ice at temperature $T$, from Bitz and Lipscomb (1999) is
$$q(S,T) = \rho c_0 (T_m - T) + \rho q_{fresh}\left(1 + \frac{\mu S}{T}\right). \qquad (5)$$

At the lower surface of the ice, $q$ is calculated by setting $T = -1.8°C$ and $S = 4.0$ as above. At the upper surface of the ice, $T$ is usually extremely close to 0°C when melting is taking place, meaning that a choice of $S$ that is both consistent and physically realistic in all cases is difficult to make. Instead, it is assumed that the ice at the upper surface is fresh, and $q = q_{fresh}$ is used.

The monthly heat fluxes calculated above are subject to several sources of uncertainty. These are evaluated in detail in Section 3.3 below, but the issues are briefly summarised here. Firstly, there is significant uncertainty due to lack of knowledge of ice salinity, which affects the fluxes through the ice conductivity and heat capacity. Secondly, the manner of dependence of ice conductivity on salinity is also subject to uncertainty, with an alternative formulation to Maykut and Untersteiner being proposed by Pringle (2006). Thirdly, both snow and ice density are subject to uncertainty, affecting the diagnosis of melting and freezing fluxes at the top and basal surfaces of the ice from elevation changes (as well as sensible heat uptake in the lowest layer of the ice). Finally, the reference layers chosen to evaluate conductive and heat uptake fluxes are themselves a parameter of the analysis, and as such represent an additional source of uncertainty.

Examination of the monthly mean energy fluxes reveals several ways in which unrealistic estimates might be produced. Firstly, in a small minority of months top or basal ice temperature is warmer than the melting point associated with the assumed salinity (1 at the top of the ice and 4 at the base) resulting in the conduction or sensible heat uptake being very large or undefined. For these months, the salinity is set instead to the highest physically allowable value, given the maximum temperature attained.

A second problem relates to the formation of false bottoms under sea ice, as documented by Notz (2003), in which meltwater refreezes upon meeting cold seawater at a temperature below its own melting point. This process visibly occurs during the period of operation of some buoys (for example 2015A, demonstrated in Figure S1), associated with sudden step changes in base elevation. These result in very large negative monthly mean ocean heat fluxes being calculated during the month of formation, and correspondingly large positive fluxes during the month of dissipation. These fluxes are physically unrealistic, as the large changes in elevation usually represent the freezing and melting of only a very thin layer of ice, with liquid seawater remaining in between this layer and the main body of the ice column. In some cases, it may be possible to estimate true ocean-to-ice heat flux simply by interpolating base elevation between the apparent times of formation and dissipation, but this approach is likely to be inaccurate for long-lived false bottoms. For the purposes of this study all affected ocean heat fluxes were simply removed from the dataset, as they were relatively few in number.

## 3. Description of monthly mean flux distributions from the IMBs

### 3.1 Seasonal and spatial variability

Throughout the description of the IMB-estimated fluxes here, and the model evaluation below, the convention used is that positive numbers denote downwards fluxes, and vice versa. The distributions of monthly mean fluxes of top melting, top conduction, basal conduction and ocean heat flux are summarised in Table 1. The IMBs provide 463 monthly mean values of top melt in total, ranging from 31 values in March and August to 53 in May. The

seasonal cycle reaches its maximum in July, when top melting of $29.9 \pm 17.8$ Wm$^{-2}$ is observed. Strong top melting is also evident in June ($16.8 \pm 11.0$ Wm$^{-2}$), but top melting tends to be considerably lower in August ($8.1 \pm 6.7$ Wm$^{-2}$). In all three summer months, the distribution is positively skewed, with a small number of very high values (for example, the highest top melt value recorded is 79.9 Wm$^{-2}$, for the buoy 1993A in July 1993). Values for the rest of the year are zero or near-zero. Throughout the year, standard deviation of the distributions is of a similar order of magnitude to the mean, showing a high degree of spatial and interannual variability.

Top conductive flux, a component of the surface energy balance, is the means by which the ice loses energy to the atmosphere in the presence of atmospheric cooling during the Arctic winter. It depends strongly upon atmospheric conditions, but also upon ice and snow thickness, as thinner ice and snow can support stronger temperature gradients and conduct energy upwards more quickly. For the top conductive fluxes, the IMBs provide 414 estimates in total, ranging from 24 in August to 51 in May. Mean top conductive fluxes are strongly negative from October-March, reaching a minimum value of $-17.6 \pm 6.8$ Wm$^{-2}$ in December. However, values are weakly positive in June and July, reflecting warming of the ice interior.

The basal conductive flux acts to remove energy from the ice base in winter, allowing ice growth, and to a lesser extent during late spring and early summer while the ice is warming, attenuating ice melt. For the basal conductive fluxes the IMBs provide 463 estimates, ranging from 29 in August to 52 in May. The basal conductive flux displays a seasonal cycle less amplified than, and displaced slightly later relative to, that of the top conductive flux, with lowest values occurring from November-April and a minimum of $-14.0 \pm 5.7$ Wm$^{-2}$ occurring in January. The damped response relative to the top conductive flux occurs due to the thermal inertia of sea ice, and the principal thermodynamic forcing occurring at the top surface.

Lastly, for the ocean heat fluxes the IMBs provide 414 estimates, ranging from 25 in August to 49 in May. The highest values are seen in July and August, with a mean and spread of $18.1 \pm 15.3$ Wm$^{-2}$, and $19.2 \pm 23.9$ Wm$^{-2}$ respectively. The distributions in these months are, like the top melting flux, strongly positively skewed, with a small number of exceptionally high values. Notably, 119 Wm$^{-2}$ is estimated in August 2007 for the buoy 2006C in the Beaufort Sea, as part of a summer of extreme ice melt documented by Perovich et al. (2008). In the winter, mean values of ocean heat flux are near-zero. There is frequent occurrence of small negative estimates in the distributions in the winter. These are likely to be spurious and reflect errors in assumptions made about the salinity and density at the base of the ice. For most such values, the uncertainty interval resulting from varying the salinity from 0 to 10 encompasses 0 Wm$^{-2}$.

Two regions of the Arctic are relatively densely sampled by the IMBs: the Beaufort Sea and the North Pole (Figure 3). In order to demonstrate that the IMBs are able to capture some regional variability, and especially to aid with model evaluation in Section 4 below, monthly mean fluxes derived from buoy tracks entirely within these regions were sorted into separate datasets, characteristics of which are now described separately. Mean and standard deviations of the distributions in the North Pole and Beaufort Sea regions are summarised in Tables 2 and 3 respectively; boxplots are presented in Figure 7. Significance of differences between distributions is measured using a Welch t-test, with a 5% p-value threshold.

Top melting fluxes are shown in Figure 7a separately for the Beaufort Sea and the North Pole regions. In June, the top melting fluxes measured in the North Pole region range from 1 to 37 $Wm^{-2}$, with a mean of $12 \pm 8$ $Wm^{-2}$, while those measured in the Beaufort Sea range from 10 to 52 $Wm^{-2}$ with a mean of $26 \pm 10$ $Wm^{-2}$. The lower distribution in the North Pole region is consistent with the observed later onset of surface melting here (Markus et al., 2009) associated with the higher latitude. In July, measured fluxes range from 2 to 55 $Wm^{-2}$ in the North Pole region, with a mean of $23 \pm 14$ $Wm^{-2}$, and 11 to 80 $Wm^{-2}$ in the Beaufort Sea region, with a mean of $41 \pm 17$ $Wm^{-2}$. In both June and July, distributions of top melt fluxes are significantly different in the two regions. Measured fluxes of top melting are much lower in August in both regions.

For the top conductive flux (Figure 7b), winter fluxes tend to be slightly higher in magnitude in the North Pole than in the Beaufort Sea region, although in no winter months are the distributions significantly different at the 5% level. In January, for example, North Pole fluxes range from -32 to -10 $Wm^{-2}$ with a mean of $-18 \pm 7$ $Wm^{-2}$, while those in the Beaufort Sea region range from -20 to -7 $Wm^{-2}$ with a mean of $-15 \pm 7$ $Wm^{-2}$. Some notable differences between the distributions occur in the 'shoulder seasons', particularly in May and August (when the distributions are significantly different), with higher values, indicating ice warming, occurring in the Beaufort Sea region. For example, in May, values in the North Pole region range from -6 to 3 $Wm^{-2}$ with a mean of $-1 \pm 2$ $Wm^{-2}$, while values in the Beaufort Sea region range from -2 to 4 $Wm^{-2}$ with a mean of $1 \pm 2$ $Wm^{-2}$. These differences indicate earlier onset of warming in the Beaufort Sea and earlier onset of cooling in the North Pole region, consistent with an earlier onset of surface melt in the Beaufort Sea.

Less spatial variability is evident for the mean basal conductive flux (Figure 7c). For example, in December, North Pole fluxes range from -20 to -7 $Wm^{-2}$ with a mean of $-13 \pm 3$ $Wm^{-2}$, while Beaufort Sea fluxes range from -32 to -1 $Wm^{-2}$ with a mean of $-14 \pm 7$ $Wm^{-2}$. Hence the thermal inertia of ice appears to have some damping effect on the larger variability of thermal forcing evident in the Beaufort Sea region from the top conductive flux. Winter variability tends to be higher in the Beaufort Sea than the North Pole, but this is largely caused by a small number of exceptionally low fluxes early in the winter associated with end-of-summer ice thicknesses of 50cm or lower, notably a value of -61.7 $Wm^{-2}$ recorded in October 2007 for the buoy 2006C. The faster warming and slower cooling of ice evident in the shoulder seasons in the Beaufort Sea region for the top conductive flux is also not evident for the basal conductive flux. In the month of May, for example, basal conductive flux values range from -13 to 0 $Wm^{-2}$ in the North Pole region with a mean of $-7 \pm 3$ $Wm^{-2}$, compared to a range of -8 to -2 $Wm^{-2}$ and a mean of $-5 \pm 1$ $Wm^{-2}$ in the Beaufort Sea region.

For the ocean heat flux (Figure 7d), in the summer very high values tend to be more common in the Beaufort Sea region than in the North Pole region. For example, in August North Pole region values range from 2 to 38 $Wm^{-2}$ with a mean of $13 \pm 10$ $Wm^{-2}$, while the Beaufort Sea region values range from 7 to 119 $Wm^{-2}$ with a mean of $33 \pm 35$ $Wm^{-2}$. It is likely that these are related to the lower ice fractions, and greater solar heating of the mixed layer, in the Beaufort Sea region.

## 3.2 Interannual variability

Having examined spatial and seasonal variability in the estimated fluxes, it is natural to consider whether the dataset also gives useful information about interannual variability. Restricting fluxes to individual years does not give enough data points (per month) to permit analysis, particularly early in the period where very often there was only one or two buoys in operation in any particular year (and in some cases none). Instead, the period of the IMB operations is divided into three sections with very roughly equal numbers of data points for each flux and month: 1993-2006, 2007-2012 and 2013-2015. The middle period is chosen to contain the two years with the lowest September extents (2007 and 2012, according to HadISST1.2) in the entire analysis period, to maximise the chance that interannual variability can be detected. For each flux, and month of year, we compare the distribution of values estimated for each period (Figure 8) and use a Welch t-test to judge whether any distributions are significantly different at the 5% level.

For the top melting flux, in no months are the distributions significantly different, and in July and August means and standard deviations are very similar. In May, however, the mean top melting flux is higher in the later period 2013-2015 ($2.7 \pm 5.8$ Wm$^{-2}$) than in the middle ($0.5 \pm 1.$ Wm$^{-2}$) and early ($0.5 \pm 0.9$ Wm$^{-2}$) periods. By contrast, in June the mean top melting flux is lower in the later period ($13.3 \pm 9.6$ Wm$^{-2}$) than in the middle ($19.0 \pm 8.8$ Wm$^{-2}$) and early (13.7 Wm$^{-2}$) periods. The differences could conceivably reflect the observed trend towards earlier onset of melt (e.g. Bliss et al, 2019), but the lack of significance makes drawing conclusions difficult.

Early in the winter, the top conductive flux becomes higher (less negative) as time passes: for example, in October the distribution means are -17.9 $\pm$ 12.5, -13.2 $\pm$ 5.1 and -11.9 $\pm$ 5.6 Wm$^{-2}$ for the periods 1993-2006, 2007-2012 and 2013-2015 respectively. Late in the winter, the trend reverses, with top conductive flux becoming lower (more negative) as time passes: for example, in March the distribution means are -10.8 $\pm$ 4.1, -14.2 $\pm$ 5.9, -14.6 $\pm$ 4.2 Wm$^{-2}$ for the three periods respectively. However, as with the top melting fluxes the distributions are not significantly different. Difference in basal conductive flux are still less marked, with distributions similar in all months except October, where the middle period displays a much higher (more negative) mean and greater spread due to the presence of a small number of extreme values (including that from the buoy 2006C, noted above).

For the ocean heat flux, an upward trend is apparent in the month of July, with means of 10.1 $\pm$ 4.4, 20.0 $\pm$ 17.0 and 23.6 $\pm$ 17.1 Wm$^{-2}$ for the three periods respectively: for the early and late periods, the distributions are barely significantly different. In August, the mean in the middle period ($32.5 \pm 37.0$ Wm$^{-2}$) is much higher than that of the early ($14.0 \pm 15.1$ Wm$^{-2}$) and late ($14.1 \pm 9.5$ Wm$^{-2}$) periods, but the differences are not significant. The paucity of significant differences between distributions, coupled with the deliberate choice of periods to maximise interannual variability, suggests that it is difficult to detect robust interannual trends in the IMB dataset in its current state.

### 3.3 Uncertainty associated with assumptions of the analysis

We assess uncertainty due to ice salinity, snow and ice density, ice conductivity and to the layers used to calculate conductive flux and ocean heat flux. Guided by estimates produced in the modelling studies of Turner et al. (2015)

and Vancoppenolle et al. (2008), we use a practical salinity range of 0 – 10 to evaluate uncertainty due to salinity
at both upper and basal surfaces of the ice. In fact, the ice salinity causes by far the greatest uncertainty in all
measured fluxes, and the effect is most marked when considering the top melting flux. For example, the top
melting flux estimated from the buoy 1997D in the month of July 1998 is 31.0 $Wm^{-2}$ when a salinity of 0 is
assumed; but 0.4 $Wm^{-2}$ with a salinity of 10. This is due to the much lower latent heat of fusion of ice at higher
salinities. Over the distribution of a whole, average July top melting flux is 29.9 $Wm^{-2}$ with a salinity of 0 but 1.6
$Wm^{-2}$ with a salinity of 10.
At first sight, the large uncertainties would render evaluation of the top melting flux in a sea ice model using IMB
data extremely difficult. However, the physical meaning of this uncertainty must be correctly understood. The
specific latent heat of high salinity ice is higher because a significant fraction of the ice will already have
undergone melting. The energy used in melting this ice is accounted for in sensible heating of the top layer of ice,
as high salinity ice has a higher heat capacity for this reason. In a sense, top melting of ice, and sensible heating
of the top layer, are part of the same process. Undertaking a meaningful evaluation of modelled top melting using
the IMB fluxes therefore requires consideration of the thermodynamic treatment of ice in that model. For example,
in a model such as HadGEM2-ES, it is appropriate to compare modelled top melting to energy used in melting
the entire top layer of ice – equivalent to assuming an ice salinity of 0 in the IMB dataset. This is because
HadGEM2-ES does not model ice salinity or heat capacity (as described in more detail in Section 4 below).
The salinity has a much smaller, though still noticeable, effect on the conductive flux. In February 2014, for
example, a salinity of 0 is associated with a top conductive flux of -12.5 $Wm^{-2}$, while a salinity of 10 is associated
with a flux of -11.8 $Wm^{-2}$. Over the whole dataset, the average February top conductive flux is -17.0 (-16.6) $Wm^{-2}$
$^{2}$ when a salinity of 0 (10) is assumed. Sensitivity is higher in the summer, as conductivity is more sensitive to
salinity at higher temperatures: over the dataset, the average July top conductive flux is 3.1 (-0.1) $Wm^{-2}$ when a
salinity of 0 (10) is used. The basal conductive flux displays highest sensitivity to salinity from February – April:
for example, the average March basal conductive flux is -13.3 (-11.7) $Wm^{-2}$ when a salinity of 0 (10) is assumed.
Ocean heat fluxes tend to display higher sensitivity to salinity than do the conductive fluxes, but lower than does
the top melting flux. This is mainly because temperatures tend to be lower at the basal surface of the ice than at
the top during the summer (when top melting and ocean heat fluxes tend to be greatest in magnitude), reducing
sensitivity of the latent heat of fusion of ice to salinity. For example, in August 2003 the buoy 2003D displays an
ocean heat flux of 24.3 (16.6 $Wm^{-2}$) when salinity of 0 (10) is assumed. For the distribution as a whole sensitivity
is highest in the month of August when the mean ocean heat flux is 23.0 (13.5) $Wm^{-2}$ when salinity of 0 (10) is
assumed.
To examine sensitivity to snow density, we use the range 274-374 $kgm^{-3}$, after Alexandrov et al (2010). Snow
density only affects the top melting flux: the highest sensitivity is seen in the month of June, where the average
top melting flux is 15.4 (17.9) when snow density of 274 (374) $kgm^{-3}$ is used. We also examine sensitivity to ice
density, using the range 917-944 kgm-3, after Cox and Weeks (1982): for the top melting flux, the highest
sensitivity is in July, when the average top melting flux is 29.9 (30.7) $Wm^{-2}$ when ice density is 917 (944) $kgm^{-3}$.

The ocean heat flux also depends on ice density, and the largest difference occurs in the month of August, when the average flux is 19.9 (20.5) Wm$^{-2}$ when ice density is 917 (944) kgm$^{-3}$.

Ice conductivity is also subject to considerable uncertainty. An alternative formulation to the Maykut and Untersteiner method used in this study was proposed by Pringle (2006) following laboratory tests of land-fast sea ice, in which sea ice conductivity $k_I$ is calculated from ice temperature $T$ (in °C) and practical salinity $S$ as

$$k_I = 2.11 - 0.011T + 0.09\frac{S}{T} \qquad (6)$$

Sensitivity of the IMB-measured fluxes to the conductivity formulation was tested by recalculating conductive and ocean heat fluxes using this alternative method (there is no difference in the top melting fluxes by design). Large difference in the winter top conductive fluxes are apparent, due to the Pringle formulation tending to produce much higher conductivities at low temperatures. For example, for the buoy 1993A in January 1994 a top conductive flux of -18.3 Wm$^{-2}$ is estimated using the Pringle formulation, but only -15.8 Wm$^{-2}$ using the Maykut and Untersteiner formulation. For the dataset as a whole, a mean January top conductive flux of -21.0 Wm$^{-2}$ is estimated with the Pringle formulation and -17.7 Wm$^{-2}$ with the Maykut and Untersteiner formulation.

Finally, sensitivity of the IMB basal conductive and ocean heat fluxes to the depth and thickness of the reference layers used was tested. The fluxes were recalculated with the lowest 20cm of the ice used to calculate sensible heat uptake, and the layer 20-40cm above the ice base to calculate basal conductive fluxes. The largest change in mean basal conductive flux occurs in October, with a mean value of -0.7 Wm$^{-2}$ as opposed to -4.1 Wm$^{-2}$ in the standard configuration. This is associated with temperature gradients being smaller closer to the ice base. The difference decreases through the winter, with -11.7 Wm$^{-2}$ in February, as opposed to -13.7 Wm$^{-2}$ in the standard configuration. The largest difference in ocean heat flux also occurs in October, with a mean value of 2.8 Wm$^{-2}$ as opposed to 5.4 Wm$^{-2}$ in the standard configuration.

In summary, varying parameters of the analysis results in measurable changes to the IMB fluxes. In most cases however, the sensitivity of the fluxes to the parameters is an order of magnitude lower than the absolute values, in the months of the year when the absolute values tend to be at their peak (winter for the conductive fluxes, summer for the top melting and ocean heat fluxes). The main exception is the effect of salinity on the top melting fluxes in summer, but as noted above care is needed when interpreting this uncertainty in the context of a model evaluation.

## 4. Evaluating modelled sea ice using the IMB-derived fluxes
### 4.1 Evaluating vertical energy fluxes in HadGEM2-ES with the IMBs

In this section, the distributions of energy fluxes estimated from the IMB data are compared to equivalent fluxes simulated by the coupled climate model HadGEM2-ES. This model, developed from the earlier model HadGEM1, is based on the HadGEM2-AO coupled atmosphere-ocean system, but employs additional components to simulate terrestrial and oceanic ecosystems, and tropospheric chemistry (Collins et al., 2011). The sea ice component is

very similar to that used by HadGEM1 (McLaren et al., 2006), employing a subgrid-scale thickness distribution with 5 categories (Thorndike et al., 1975), elastic-viscous-plastic rheology (Hunke and Dukowicz, 1997), and a zero-layer thermodynamics scheme, described in the appendix to Semtner (1979), in which sea ice has no heat capacity and conduction does not vary with height within the ice. The atmosphere and ocean components contain a number of improvements relative to HadGEM1 (Martin et al., 2011). The sea ice in HadGEM2-ES is modelled on a regular latitude-longitude grid, with a resolution of 1 degree throughout the Arctic.

The Arctic sea ice and surface radiation simulation of the historical ensemble of HadGEM2-ES was evaluated by West et al (2019). A number of likely model biases were identified; a low bias in September sea ice extent, a low bias in annual mean ice thickness and a high bias in ice thickness seasonal cycle amplitude, and a tendency to model overly high surface net downwelling shortwave (SW) flux in summer and overly low surface net downwelling longwave (LW) flux in winter. To perform the evaluation of the ice energy budget in the present study, the model period 1980-1999 is used, chosen to match the period used in West et al (2019).

For each month, grid cells lying inside the North Pole and Beaufort Sea regions (Figure 3) are separately identified, and monthly mean top melt flux, top conductive flux, basal conductive flux and ocean heat flux are collected into distributions. Means and standard deviations of these distributions are then compared to those of the IMB fluxes, with model fluxes weighted by grid cell area when calculating these statistics. Before aggregation the fluxes, produced by the model as grid-box means, are divided by ice area to produce means over ice only, for greater consistency with the IMB fluxes. As above, to compare modelled and observed distributions in a systematic manner, a Welch t-test is used, and the 5% level is chosen as a threshold for significance of difference of distributions.

The approach of calculating model distributions over whole regions and time periods, rather than comparing only grid cells lying under IMB tracks during the respective month, is chosen for two reasons. Firstly, internal variability is such that no climate model would be expected to capture the exact atmospheric conditions over a specific IMB track; whether the model could capture the average conditions over a larger-scale region and time period is a more useful question for evaluation purposes. Secondly, even a model grid-cell implicitly models sea ice of many different thicknesses, due to the ice thickness distribution, and therefore the comparison with IMB tracks cannot be made more like-for-like by restricting to a single grid cell.

For both regions, top melt fluxes simulated by HadGEM2-ES tend to be much higher than those measured by the IMBs (Figure 9a,b): modelled and observed distributions are significantly different throughout the melt season. For example, the modelled mean top melt flux of $72.5 \pm 8.2$ Wm$^{-2}$ in the Beaufort Sea region in June is much higher than the IMB mean of $26.0 \pm 10.2$ Wm$^{-2}$; in the North Pole region in June, the modelled mean top melt flux of $56.6 \pm 14.0$ Wm$^{-2}$ is much higher than the IMB mean of $12.4 \pm 8.3$ Wm$^{-2}$. A Welch t-test shows the modelled and observed distributions of top melt fluxes to be significantly different at the 5% level throughout the summer in both regions. The phase of the annual cycle in top melt is shifted slightly earlier, with the effect that, in both regions, the modelled June and July means are very similar while the IMB estimates show a distinct maximum in July. However, the greater top melt in the Beaufort Sea region relative to the North Pole region is captured by the model. The bias towards excessive top melting displayed by HadGEM2-ES is consistent with the

finding by West et al (2019) that summer net SW fluxes are overestimated in HadGEM2-ES. It was shown in this study that this was likely associated with an early onset of surface melting in the model. In turn, this triggers the meltpond parameterisation of HadGEM2-ES, lowering surface albedo at an earlier time of year than meltponds would have formed in reality.

The annual cycle of top conductive flux is broadly captured by HadGEM2-ES (Figure 9c,d), with strongly negative values modelled in the autumn, winter and spring and weakly positive values in the summer. However, from September-May modelled means are more strongly negative than observed means: for example, in the North Pole region in December a modelled mean of $-31.0 \pm 7.6$ Wm$^{-2}$ is higher in magnitude than the IMB mean of -$18.1 \pm 12.7$ Wm$^{-2}$. Ice thickness in HadGEM2-ES is known to be biased low in early winter (West et al, 2019), and in section 4.2 below the extent to which this bias is responsible for the conductive flux bias is investigated. The higher magnitude of conductive fluxes in winter in the Beaufort Sea region relative to the North Pole region is captured by the model; however, the higher values of fluxes in May and September in the Beaufort Sea region relative to the North Pole region are not captured. Modelled and observed flux distributions are significantly different in all months except February, March and November (North Pole region) and June, September and October (Beaufort Sea region).

Modelled values of basal conductive flux at each model grid-box are identical to those of top conductive flux, due to the HadGEM2-ES zero-layer thermodynamics scheme. Consequently HadGEM2-ES overestimates the magnitude of basal conductive flux in autumn and winter more severely than it does top conductive flux (Figure 9e,f). The overestimation is most severe during the autumn; in the Beaufort Sea region in October a mean modelled flux of $-28.1 \pm 11.1$ Wm$^{-2}$ is much higher in magnitude than the mean observed flux of $-7.9 \pm 16.8$ Wm$^{-2}$. As the basal conductive flux in the freezing season is the principal driver of ice growth, this suggests that HadGEM2-ES is likely to model substantially stronger ice growth during these months than was measured at the IMB sites. Modelled and observed fluxes are significantly different in all months and regions except, barely, in the North Pole region in August.

For the ocean heat flux, in the Beaufort Sea region the model produces a similar seasonal cycle to that estimated from the IMB data, with very small values in the winter and a wide range of positive values in the summer (Figure 9g,h); only in April and May are the distributions significantly different. For the North Pole region however, two differences are apparent. Firstly, spread in winter ocean heat fluxes is much higher in the model, with a small number of very high fluxes; these occur near the southern boundary of the region, where the ice cover meets warmer water moving north through the Fram Strait. Secondly, modelled ocean heat fluxes in summer tend to be higher than those measured by the IMBs. For example, in July a modelled mean of $27.2 \pm 13.9$ Wm$^{-2}$ compares to an observed mean of $12.1 \pm 7.4$ Wm$^{-2}$. In the North Pole region, modelled and observed distributions are significantly different in all months except September.

The HadGEM2-ES ocean heat flux distributions are in fact similar between the North Pole and Beaufort Sea regions, but evaluate differently because the IMB ocean heat fluxes in the North Pole region are much lower. In other words, the spatial variability in summer ocean heat fluxes suggested by the IMB data is not captured by the model. This may be related to late summer ice concentration biases in HadGEM2-ES: ice concentration is biased

low in the North Pole region, but not in the Beaufort Sea region. This would tend to cause a high bias in net SW absorption by the ocean mixed layer, and hence a positive bias in ocean heat flux.

It is instructive also to examine how modelled oceanic heat convergence (OHC) compares to real-world estimates. Arctic Ocean heat convergence in HadGEM2-ES from 1980-1999 is 4.9 $Wm^{-2}$, roughly consistent with estimates of heat transport through the Fram Strait, the dominant pathway by which the Arctic Ocean gains heat (e.g. Schauer et al, 2004). However, oceanic heat convergence in the North Pole region (as defined in Figure 3) is higher at 9.1 $Wm^{-2}$, while oceanic heat convergence in the Beaufort Sea region is lower at 1.6 $Wm^{-2}$. These patterns are consistent with most of the Arctic Ocean heat convergence being released relatively close to the Fram Strait and Barents Sea, the principal points of ingress of relatively warm Atlantic water. It is noteworthy that the large difference in OHC between the two regions is not mirrored in the ocean-to-ice heat flux. This is consistent with the finding by Keen et al. (2018) that much of the ocean-to-ice heat flux in HadGEM2-ES derives from direct solar heating, rather than OHC. Studies by Bitz et al. (2008), Perovich et al. (2008) and Steele et al. (2010) show this is likely to be true in the real world also. Hence differences in OHC are unlikely to contribute to differences in ocean-ice heat flux.

The model biases in summer top melt and winter conductive fluxes are larger than most of the uncertainties measured in section 3.3. The model bias in top melt is of a similar magnitude to the uncertainty in top melting due to salinity, albeit in the opposite direction (higher salinities imply an even greater model bias). However, for the reasons stated in section 3.3, the most meaningful comparison is obtained by considering the energy used in melting the whole of the top layer of ice, including the melting of brine pockets during sensible heating – equivalent to using a salinity of 0 in calculating top melting. Hence the model biases stated above – for the 'standard configuration' – are likely to present the most accurate picture.

## 4.2 Links to sea ice and surface radiation simulation of HadGEM2-ES

A top melting bias of ~40 $Wm^{-2}$ is estimated for the month of June in both the North Pole and Beaufort Sea regions. This is consistent with the finding of West et al (2019) that June surface net shortwave (SW) radiation in the model was biased high relative to a variety of satellite and reanalysis datasets, by around 20 $Wm^{-2}$ over the Arctic Ocean on average. Relative to CERES-EBAF measurements from 2000-2013 (Loeb et al., 2009), HadGEM2-ES overestimates June net SW in the North Pole and Beaufort Sea regions by 30 and 9 $Wm^{-2}$ respectively. Owing to the recent trend to earlier onset of surface melt over the past 30 years (e.g. Markus et al., 2009), and attendant likely decrease in surface albedo, these biases are likely to be underestimated. Hence it is likely that a major part of the model bias in top melting can be explained by a model bias in net SW. In West et al (2019) it was shown that a tendency for surface melt onset to occur too early in HadGEM2-ES, reducing the surface albedo in a parameterisation of the effect of melt-ponds, was likely to be principally responsible for this.

The severe overestimation in magnitude of basal conductive fluxes during the early part of the melt season can be partly explained by the zero-layer thermodynamics scheme of HadGEM2-ES; the thermal inertia effect seen in

the IMBs, whereby the basal conductive flux drops much more slowly in autumn than the top conductive flux, does not occur in the model. However, as the top conductive fluxes also tend to be considerably higher in the model than in the IMB estimates, the thermal inertia effect is likely to be only partially responsible. In West et al (2018) two other model biases were identified as being likely to lead to a high bias in winter ice growth (analogous to the basal conductive flux bias): a negative bias in ice thickness during early winter, and a negative bias in downwelling longwave (LW) radiation throughout the season. It was estimated that these biases were likely to lead to surface flux biases of order ~10 Wm$^{-2}$ throughout the freezing season. Hence these are also likely to explain a portion of the basal conductive flux bias noted above.

The excessive modelled top melting and basal conductive fluxes identified would be likely to lead to too strong ice growth, and ice melting, in winter and summer respectively, and an associated amplification of the ice thickness seasonal cycle. Such an amplification was identified in HadGEM2-ES by comparing modelled ice thickness to the forced ice-ocean model PIOMAS, as well as to estimates from satellites and submarines. Hence there is a high level of consistency between the model biases inferred from the IMB estimates, and those inferred from the sea ice state and surface radiation evaluations in West et al (2018). The IMB evaluation, however, provides additional insight to the picture, by providing consistent evaluation of previously unknown processes such as top melting. In addition the IMB evaluation clearly identifies a role for the zero-layer thermodynamics scheme in driving a bias towards excess ice growth during the early winter in HadGEM2-ES.

### 4.3 The relationship between conductive flux, ice thickness and snow depth

Both top and basal conductive fluxes are strongly related to ice thickness, as thicker ice tends to have weaker temperature gradients than thinner ice under similar atmospheric conditions. Conductive fluxes are also related to snow depth for similar reasons. To examine the extent to which the HadGEM2-ES conductive flux biases are influenced by biases in ice thickness, conductive fluxes from November – March were aggregated into ice thickness bins, 20cm wide and ranging from 0-4m, in both IMBs and model. HadGEM2-ES ice thicknesses are much lower than ice thicknesses sampled by the IMBs from November – March in both the North Pole and Beaufort Sea regions, with most of the 'overlap' occurring in the range 1-2m (Figure 10).

For the North Pole region, IMB-measured top conductive flux is similar in magnitude to HadGEM2-ES conductive flux for the overlapping range of thicknesses, but IMB-measured basal conductive flux tends to be lower. For example, in the 1.4-16m range the IMB-measured top and basal conductive flux range from -33.0 to -11.9 Wm$^{-2}$ and -24.0 to -10.8 Wm$^{-2}$ respectively, while the HadGEM2-ES conductive flux range from -15,6 to -26.6 Wm$^{-2}$. As above, this suggests that the uniform conductive flux assumption of HadGEM2-ES is important in driving excessive ice growth in this model. However, in the Beaufort Sea region both top and basal conductive fluxes from the IMBs are much lower than HadGEM2-ES even in the region of overlapping thicknesses. For example, in the 1.4-1.6m range the IMB-measured top and basal conductive flux range from -18.8 to -7.8 Wm$^{-2}$ and -18.4 to -8.3 Wm$^{-2}$ respectively, while the HadGEM2-ES conductive flux ranges from -30.0 to -21.4 Wm$^{-2}$.

This indicates that in the Beaufort Sea at least, ice thickness biases and the uniform conductive flux assumption
are not the only factors driving excessive ice growth: biases in atmospheric thermal forcing are also at work.
The HadGEM2-ES conductive fluxes display an inverse relationship with ice thickness. Cells with thinner ice
tend to have higher conductive flux (Figure 10a-b): in the Beaufort Sea region for example, the correlation
coefficient between conductive flux and the logarithm of ice thickness is 0.66. Curiously, there is no sign of a
similar relationship in the IMBs in either region; in the Beaufort Sea region the correlation coefficient between
the log of ice thickness and top (basal) conductive flux is small and not significant at 0.25 (-0.16). In particular,
both regions exhibit large numbers of IMB measurements with high ice thicknesses and high top conductive
fluxes. In most cases these points are associated with strong ice cooling and a very low basal conductive flux.
When conductive flux is compared to snow depth, it can be seen that similar snow depths are associated with
much greater conductive fluxes in HadGEM2-ES than in the IMB data (Figure 10c-d), indicating that snow depth
biases are unlikely to be a major contributor to the conductive flux biases. A similar inverse relationship between
conductive flux and snow depth is seen in HadGEM2-ES, stronger in the Beaufort Sea than at the North Pole.
Unlike with the ice thickness, there is the suggestion of a similar relationship in the IMB data, with the 10-15cm
(5-10cm) category in the North Pole (Beaufort Sea) displaying much higher conductive fluxes than are present in
the other categories. No data points with high snow depths display high conductive fluxes. As a result, the
correlation between top (basal) conductive flux and the log of ice thickness in the IMB estimates is 0.47 (0.48).
**5.    Representativeness of the IMB-estimated fluxes**
The comparison of HadGEM2-ES modelled fluxes to those inferred from the IMB measurements reveals several
potential model biases, notably the overestimation of top melt flux in June and July, and overestimation of the
magnitude of basal conductive flux in the early freezing season. In this section the accuracy of this method of
model bias estimation is discussed.
The model flux distributions evaluated represent area-weighted means over the ice-covered fractions of grid cells,
each of order 10-100 km in width, chosen to cover the North Pole and Beaufort Sea regions as defined in Section
2. The 'true model bias' would represent the difference between this and the average flux over ice covered regions
for each month in the 1980-1999 period. By contrast, the IMB-measured means are derived from a relatively small
number of single point measurements from these regions (points that tend to move position during an individual
month, with the general ice flow), most from a period somewhat later than 1980-1999. To assess the accuracy of
the model biases inferred, a method of estimating the order of magnitude of the likely error in the IMB estimated
mean fluxes is required.
One source of error derives from the temporal offset in the IMB measurements relative to the model. To assess
the impact of this, we compare the HadGEM2-ES vertical ice fluxes from the period 1980-1999 to those from the
period 2000-2015, during which most of the IMB measurements were taken (as the historical experiments end in
2005, the RCP8.5 experiment was used from 2006 onwards). Flux anomalies in the later period relative to the

earlier period are mostly small (below 2 $Wm^{-2}$ in magnitude) in both regions. However, there is a significant negative anomaly in July top melting, -6 $Wm^{-2}$ and -9 $Wm^{-2}$ in the North Pole and Beaufort Sea regions respectively. There is also a positive anomaly in September basal conductive flux (3 $Wm^{-2}$ in both regions), and in the Beaufort Sea moderate basal conductive flux anomalies continue into the winter, being 5, 3, -4 and -2 $Wm^{-2}$ from October-January respectively. These anomalies are likely to reflect earlier melting and later freezing of ice in this region. They are small in size compared to the HadGEM2-ES model biases, but suggest that the temporal bias may cause model top melting bias in July, and Beaufort Sea basal conductive flux in autumn, to be slightly overstated.

A potentially more serious source of error is sampling: bias would be introduced if the IMB measurements were systematically over- or under-sampling locations with higher than average flux in a particular month. The Arctic sea ice cover is highly heterogeneous, with ice conditions varying substantially over all scales. For most variables (for example snow thickness, ice salinity or ice albedo), it is difficult to assess whether the variability of the ice pack is sufficiently sampled by the IMB measurements, due either to a lack of reference datasets or to an inability to estimate these variables at the IMB locations. However, the degree to which the ice thickness distribution (which affects conductive fluxes in particular) is correctly sampled by the IMBs can be assessed. The effect of errors in the ice thickness distribution on the IMB-measured fluxes is estimated in Appendix A, and is shown to be small compared to the model biases identified. We note that the ice thickness distribution sampling bias is likely to be particularly strong (relative to other variables) due to the deliberate placing of IMBs in level multiyear ice, and due to the Lagrangian movement of the IMBs combined with the generally short lifetime of thin ice floes, which tend to grow quickly in winter and melt quickly in summer. It is proposed that given the effect of this bias is weak, it is likely that the effect of other, not deliberately introduced, biases is weaker still, and that the model biases identified in Section 4 are likely to be robust features.

### 6. Conclusions

Around 500 estimates of monthly mean top melt, top conduction, basal conduction and ocean heat flux have been estimated from data measured by the Arctic IMB network, with the number of estimates available for each month ranging from 26 to 59. The distributions capture seasonal and spatial variability in the vertical fluxes analysed, but do not contain sufficient data points to capture interannual variability. Comparison of modelled fluxes to observed fluxes in the two densely sampled regions in the North Pole and Beaufort Sea reveals substantial model biases, notably to high top melt fluxes in summer and to high (negative) basal conductive fluxes in autumn and early winter. Uncertainty in the IMB fluxes due to parameters of the analysis, and biases due to inadequate sampling of thin and very thick ice types, are likely to be small relative to the model biases identified.

The flux biases are consistent with an evaluation of the sea ice simulation of HadGEM2-ES that identified an over-amplified seasonal cycle in ice thickness, with model ice growth and melt biased high in winter and summer respectively, as well as a high model bias in net SW radiation in June, a low bias in net LW radiation throughout the winter, and a low model bias in ice thickness in autumn and early winter. The IMB analysis confirms that the net SW bias is likely to cause overly strong ice thinning during summer via anomalously strong top melting of

ice. The IMB analysis also allows the effect of biases in ice thickness, snow depth, and atmospheric conditions,
as well as the effect of the uniform conductive flux assumption, on the conductive fluxes to be separately
examined. In this way it is confirmed that both low ice thickness, and cold atmospheric conditions are likely to be
driving anomalously strong winter ice growth via the basal conductive flux, both conclusions already suggested
by West et al (2019). However, the IMB analysis also suggests that the zero-layer thermodynamic scheme of
HadGEM2-ES plays a role in promoting this anomalously strong ice growth. The IMB analysis also provides
evidence that snow depth biases are not important in driving the ice growth biases of HadGEM2-ES.
The calculated IMB fluxes hence offer a valuable tool for increasing understanding of sea ice simulations in
coupled models, as they allow detailed examination of the links between atmospheric forcing of sea ice and the
resulting sea ice state. This is particularly the case for the upcoming Phase 6 of the Coupled Model
Intercomparison Project (CMIP6), for which diagnostics of ice energy and mass fluxes, such as top melting and
conduction, have been requested for all sea ice models participating in this experiment (Notz et al., 2016).
Understanding of the processes leading to biases in a given sea ice state enables better understanding of modelled
sea ice spread in the present day and in the future, and may therefore also allow better understanding of future
projections in sea ice state.
The greatest source of uncertainty in estimating the IMB fluxes derives from lack of knowledge of ice salinity at
the measurement sites, and therefore the thermodynamic properties of conductivity and heat capacity. A method
of measuring salinity at the IMB sites would greatly reduce the uncertainty in the IMB estimates, particularly for
ocean heat flux, enhancing the usefulness of this dataset as a tool for model evaluation.
**Appendix A: The ice thickness sampling bias and its effect on flux distributions**
An analysis of the distribution of monthly mean ice thicknesses sampled by the IMBs finds that most lie in the
range 1.4 – 3.6 m. However, analysis of submarine measurements of ice thickness in the Central and Western
Arctic from 1981-2000, as collated by Rothrock et al. (2008), shows a substantial proportion of ice to be of
thickness outside these bounds. In order to estimate the effect of this sampling bias, we use the following simple
model. The thickness distribution is discretised, in a similar manner to HadGEM2-ES, by separating ice into five
thickness categories, with minimum thickness bounds at 0 m, 0.6 m, 1.4 m, 2.4 m and 3.6 m. Given a mean flux
$F_{IMB}^{m,r}$ for month $m$ and region $r$ that is estimated from the IMBs by averaging all fluxes for that month and region,
the total observational error can be characterised as
$$F_{err} = F_{IMB}^{m,r} - F_{actual}^{m,r} \qquad (A1)$$
where $F_{actual}^{m,r}$ is the actual value of that flux, averaged over the region and month in question, for the period 1993-
2015. The mean flux can be further split into thickness categories by setting
$$F_{IMB}^{m,r} = \sum_i F_{IMB-cat}^{m,r,i} \bullet \left( N_i^{m,r} / N_{total}^{m,r} \right) \qquad \text{(A3)}$$
where the $F_{IMB-cat}^{m,r,i}$ are the average IMB flux over month $m$, region $r$ and category $i$, $N_i^{m,r}$ is the total number of
IMB fluxes in month $m$, region $r$ and category $i$, and $N_{total}^{m,r}$ is the total number of IMB fluxes in month $m$ and
region $r$.
Similarly the actual flux values can be written as
$$F_{actual}^{m,r} = \sum_i F_{actual-cat}^{m,r,i} \bullet a_i^{m,r} \qquad \text{(A4)}$$
where $a_i^{m,r}$ is the average fraction of ice in month $m$ and region $r$ that is in category $i$ (expressed as a proportion
of average fraction of ice in the region).
It can be seen that $N_i^{m,r} / N_{total}^{m,r}$ acts as an IMB-based estimate of $a_i^{m,r}$, and that the error in $F_{IMB}^{m,r}$ due to the
ice thickness sampling bias is exactly that due to the error in this estimate. Hence we can characterise the sampling
error by expressing $F_{err}$ in terms of systematic and sampling errors in the following way:

$$\begin{aligned}
F_{err}^{m,r} &= \sum_i \left( F_{IMB-cat}^{m,r,i} \bullet \left( N_i^{m,r} / N_{total}^{m,r} \right) - F_{actual-cat}^{m,r,i} \bullet a_i^{m,r} \right) \\
&= F_{err\_systematic} + F_{err\_sample} \\
&= \frac{1}{2} \sum_i \left( F_{IMB-cat}^{m,r,i} - F_{actual-cat}^{m,r,i} \right) \left( N_i^{m,r} / N_{total}^{m,r} + a_i^{m,r} \right) \qquad \text{(A5)} \\
&\quad + \frac{1}{2} \sum_i \left( N_i^{m,r} / N_{total}^{m,r} - a_i^{m,r} \right) \left( F_{IMB-cat}^{m,r,i} + F_{actual-cat}^{m,r,i} \right)
\end{aligned}$$

where $F_{err\_sample}$ captures the flux error due to ice thickness sampling; $F_{err\_systematic}$ describes the error due to
inaccuracy in the IMBs estimating $F$ for each particular category, effectively capturing the remaining
observational error that is beyond the scope of the current analysis.
To estimate $F_{err\_sample}$, we first need estimates of $a_i^{m,r}$, the real-world proportion of ice in a given thickness
category for each region and month, for which we use submarine measurements (National Snow and Ice Data
Center, 2019), that capture small-scale variation in sea ice thickness. Ice thickness measurements in each category
are collected for each month and region, and used to generate an estimate of $a_i^{m,r}$. In practice, measurements are
abundant during spring and autumn, but sparse during summer and non-existent during winter. To alleviate this
problem, we interpolate $a_i^{m,r}$ using a 5-month binomial mean, weighted by number of measurements, in order to
produce a smooth seasonal cycle, motivated by the observation that maximum and minimum ice thickness values
often occur during spring and autumn respectively. The resulting seasonal cycles of $a_i^{m,r}$ are shown in Figure S2.
Secondly, estimates of $\frac{1}{2}\left(F_{IMB-cat}^{m,r,i} + F_{actual-cat}^{m,r,i}\right)$, representative average fluxes for each thickness category, are
required. To calculate representative fluxes of conduction, we combine a simple model of the relationship between
conductive fluxes and ice and snow thickness with the IMB measurements of conduction. The surface flux
$F_{sfc} = F_{atmos} + BT_{sfc}$ is approximated by linearising the dependence on surface temperature $T_{sfc}$, referenced to
0°C. The surface flux is set equal to the top conductive flux $F_{condtop}$, and a constant rate of change of conductive
flux from the top to the basal surface of the ice is assumed, such that $\frac{1}{2}\left(F_{condtop} + F_{condbot}\right) = \frac{\left(T_{sfc} - T_{bot}\right)}{R_{ice}}$,
where $F_{condbot}$ represents basal conductive flux, $T_{bot}$ ice base temperature, and $R_{ice} = h_{ice}/k_{ice} + h_{snow}/k_{snow}$
is the thermal insulance of the ice-snow column, $h_{ice}$ and $h_{snow}$ being ice and snow thickness, and $k_{ice}$ and $k_{snow}$
ice and snow conductivity respectively. Eliminating $T_{sfc}$ from the above equations gives
$$F_{condtop} = \frac{F_{atmos} + BT_{bot}}{1 - BR_{ice}^{HS-top}} \qquad \text{(A6)}$$
where $R_{ice}^{HS-top} = R_{ice}/\left(1 + \alpha_{heat}\right)$ and $\alpha_{heat} = \frac{F_{condtop} - F_{condbot}}{F_{condtop} + F_{condbot}}$ .
Equation (A6) can be combined with the IMB estimates of conductive fluxes, and snow and ice thickness to
produce, for each monthly mean IMB measurement, an associated value of $F_{atmos}$, a measure of the atmospheric
forcing on the ice that is independent of small-scale variations in ice thickness. Hence for each month and region
a distribution in $F_{atmos}$ can be produced; mean values of this can be fed back into the simple model to produce
an average conductive flux that would be expected for each ice thickness category. The average conductive flux
is then multiplied by $\left(N_i^{m,r}/N_{total}^{m,r}\right) - a_i^{m,r}$, as estimated above, and summed over categories to produce the flux
error $F_{err}^{m,r}$.
The resulting flux bias is below 1 Wm$^{-2}$ in magnitude year-round in the North Pole region. It is slightly larger in
the Beaufort Sea in winter time, achieving values of -1.5 Wm$^{-2}$ to -1 Wm$^{-2}$ from November-February. The values
are small compared to the model-observation differences identified in Section 4, and so we conclude that the ice
thickness sampling bias does not seriously affect ability to evaluate modelled conductive fluxes.
A less strong, but still discernible, relationship exists between top melt flux and ice thickness, due to ice albedo
tending to be lower for thinner ice. However, this effect is likely to be associated with a significant difference in

albedo only for the thinnest category of ice, and then only in the absence of snow (e.g. Ebert et al., 1995). To estimate this effect, we use an average albedo of 0.55 for bare ice in the top four thickness categories, and 0.35 in the lowest category, based on observed values reviewed and collated by Pirazzini (2008). We assume that ice is snow-covered 80% of the time from October-May (with a corresponding albedo of 0.85), 50% of the time in June and September, and 20% of the time in July and August. Finally, we calculate mean values of downwelling SW radiation for the North Pole and Beaufort Sea region from CERES-EBAF from 2000-2013, and multiply these by the albedo differences implied by the anomaly of ice fraction in category 1. With this method, a maximum flux bias of -3 $Wm^{-2}$ is estimated for the North Pole region, in July, and of -2 $Wm^{-2}$ for the Beaufort Sea region, also in July. Again, this anomaly is small compared to the model-observation differences seen in Section 4, and it is concluded that the sampling bias similarly does not affect ability to evaluate modelled top melting.

An estimate of the influence of the sampling bias on ocean heat flux estimates is more difficult, due to a less clear relationship between ice thickness and the ocean heat flux, and the frequent presence of rapid changes in ice thickness during the months in which ocean heat flux is highest (July and August). It is likely that very small ice thicknesses are associated with elevated ocean heat flux in the summer months due to greater solar penetration through ice. However, the ice thickness sampling bias is at its least severe during the summer months, as thinner ice is sampled by the IMBs simply through the melting of originally thicker floes on which the IMBs were placed. In addition, thinning ice which induces a particularly high ocean heat flux is likely to melt out quickly, and contribute a correspondingly small fraction to the ice thickness distribution.

We examined the sensitivity of the average ocean heat fluxes to this issue by assuming ocean heat fluxes in category 1 to be systematically larger than those in the remaining categories (as diagnosed from the IMBs) by a factor $\lambda$. With $\lambda=3$, August ocean heat fluxes in the Beaufort Sea region are on average 80$Wm^{-2}$ greater in category 1 than in thicker ice categories; it is unlikely that the solar penetration effect could be associated with larger flux differences than this. The largest average flux bias associated with this effect is then -6.3 $Wm^{-2}$, also seen in August in the Beaufort Sea region. Hence this could be taken as a reasonable upper bound for the effect of the sampling bias on ocean heat fluxes. It is smaller in magnitude than the model ocean heat flux biases diagnosed, although the difference is less than those for the top melt and conductive fluxes.

**Code availability**

The code used to analyse the IMB data is published in two repositories, corresponding to two stages of the analysis. The code used to read, quality control, and process the data into consistent quantities on consistent time points, can be downloaded from https://github.com/sammiebuzzard/IMBS_MO. The code used to produce datasets of monthly mean energy fluxes from this processed data can be downloaded from https://github.com/alex-west-met-office/IMBS_Stage2_modeleval_MO. In addition, the code with which Figures 3-10, S1 and S2, were produced is published at https://zenodo.org/record/3947782#.XxAi9XdFyUk.

**Data availability**

The raw IMB data are publicly available, and can be downloaded from http://imb-crrel-dartmouth.org/results/ (Perovich et al, 2020). The processed IMB data, and the derived dataset of monthly mean fluxes, are published with this revision at https://zenodo.org/record/3773811 and https://zenodo.org/record/3773997 respectively. The diagnostics from HadGEM2-ES used in this study are not publicly available. However, they can be made available upon request to the author.

**Acknowledgements**

This work was supported by the Joint UK BEIS/Defra Met Office Hadley Centre Climate Programme (GA01 101), and the European Union's Horizon 2020 Research & Innovation programme through grant agreement No. 727862 APPLICATE. MC was supported by NE/N018486/1.

The authors thank the two anonymous reviewers for their thorough reading of the first revision of this study, and for their helpful and constructive feedback.

**Author contribution**

The IMB-derived fluxes were calculated, and the subsequent model evaluation performed, by Alex West. The analysis of sampling error in the Appendix was designed and carried out by Alex West. The paper was written in its final form by Alex West with assistance from Mat Collins and Ed Blockley.

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

| Whole Arctic | Top melt flux (Wm$^{-2}$) | | Top conductive flux (Wm$^{-2}$) | | Basal conductive flux (Wm$^{-2}$) | | Ocean heat flux (Wm$^{-2}$) | |
|---|---|---|---|---|---|---|---|---|
| Number of observations | 463 | | 414 | | 463 | | 414 | |
| | Mean | Std. dev. | Mean | Std. dev. | Mean | Std. dev. | Mean | Std. dev. |
| January | 0.0 | 0.0 | -16.2 | 6.1 | -14.0 | 5.7 | 1.4 | 5.0 |
| February | 0.0 | 0.0 | -16.9 | 6.9 | -13.7 | 6.7 | 0.6 | 4.2 |
| March | 0.0 | 0.0 | -13.5 | 5.1 | -12.7 | 4.6 | 1.5 | 5.6 |
| April | 0.0 | 0.0 | -7.5 | 3.1 | -9.7 | 3.3 | 2.3 | 2.7 |
| May | 1.1 | 3.2 | -0.5 | 2.3 | -6.2 | 2.3 | 3.4 | 4.0 |
| June | 16.8 | 11.0 | 3.8 | 1.8 | -2.2 | 1.6 | 12.3 | 16.5 |
| July | 29.9 | 17.8 | 1.0 | 1.0 | 0.5 | 1.2 | 18.1 | 15.3 |
| August | 8.1 | 6.7 | -1.1 | 3.5 | 1.0 | 1.1 | 19.2 | 23.9 |
| September | 0.6 | 1.2 | -6.3 | 4.5 | 0.7 | 1.9 | 9.4 | 11.4 |
| October | 0.0 | 0.0 | -14.4 | 8.9 | -4.0 | 11.4 | 5.4 | 13.0 |
| November | 0.0 | 0.0 | -17.3 | 7.0 | -9.2 | 9.9 | 4.6 | 7.1 |
| December | 0.0 | 0.0 | -17.6 | 6.8 | -12.5 | 6.6 | 1.3 | 5.2 |

1 **Table 1. Mean and standard deviations of fluxes measured from the IMB data in Wm$^{-2}$ in each month of**

2 **the year. For each flux, the convention is that downwards=positive.**

| North Pole region | Top melt flux (Wm$^{-2}$) | | Top conductive flux (Wm$^{-2}$) | | Basal conductive flux (Wm$^{-2}$) | | Ocean heat flux (Wm$^{-2}$) | |
|---|---|---|---|---|---|---|---|---|
| Number of observations | 196 | | 170 | | 193 | | 165 | |
| | Mean | Std. dev. | Mean | Std. dev. | Mean | Std. dev. | Mean | Std. dev. |
| January | 0.0 | 0.0 | -17.7 | 7.4 | -14.3 | 4.5 | 0.2 | 2.9 |
| February | 0.0 | 0.0 | -18.9 | 5.7 | -12.2 | 6.8 | -0.2 | 6.1 |
| March | 0.0 | 0.0 | -17.2 | 4.6 | -11.5 | 6.3 | 1.3 | 3.6 |
| April | 0.0 | 0.0 | -8.3 | 3.0 | -8.5 | 4.8 | 1.4 | 2.4 |
| May | 0.1 | 0.1 | -1.4 | 1.7 | -7.0 | 2.6 | 2.5 | 4.9 |
| June | 12.4 | 8.3 | 4.4 | 1.7 | -2.7 | 1.3 | 7.6 | 9.7 |
| July | 23.4 | 13.9 | 0.9 | 0.9 | 0.3 | 1.3 | 12.5 | 8.1 |
| August | 8.0 | 6.0 | -1.4 | 3.4 | 1.0 | 0.8 | 13.1 | 10.2 |
| September | 0.2 | 0.3 | -7.7 | 5.4 | 0.8 | 2.1 | 5.4 | 6.8 |
| October | 0.0 | 0.1 | -18.1 | 12.7 | 0.3 | 2.3 | 0.3 | 3.2 |
| November | 0.0 | 0.0 | -21.4 | 11.4 | -6.4 | 4.4 | 1.4 | 3.2 |
| December | 0.0 | 0.1 | -17.7 | 5.9 | -12.7 | 3.4 | 0.5 | 3.0 |

**Table 2. Mean and standard deviations of fluxes measured from the IMB data in the North Pole region, in Wm$^{-2}$ in each month of the year. For each flux, the convention is that downwards=positive.**

| Beaufort Sea region | Top melt flux (Wm$^{-2}$) | | Top conductive flux (Wm$^{-2}$) | | Basal conductive flux (Wm$^{-2}$) | | Ocean heat flux (Wm$^{-2}$) | |
|---|---|---|---|---|---|---|---|---|
| Number of observations | 189 | | 173 | | 202 | | 190 | |
| | Mean | Std. dev. | Mean | Std. dev. | Mean | Std. dev. | Mean | Std. dev. |
| January | 0.0 | 0.0 | -15.5 | 3.3 | -14.9 | 4.6 | 2.1 | 5.0 |
| February | 0.0 | 0.0 | -15.3 | 4.9 | -13.7 | 3.5 | 1.2 | 2.9 |
| March | 0.0 | 0.0 | -11.7 | 3.8 | -12.5 | 2.3 | 1.9 | 6.6 |
| April | 0.0 | 0.0 | -6.8 | 2.5 | -9.7 | 2.1 | 2.4 | 2.6 |
| May | 1.8 | 1.6 | 0.8 | 1.7 | -5.2 | 1.4 | 4.7 | 1.9 |
| June | 26.0 | 10.2 | 2.9 | 1.3 | -1.6 | 1.6 | 18.0 | 21.1 |
| July | 41.1 | 17.3 | 1.3 | 0.9 | 0.4 | 0.7 | 30.6 | 19.6 |
| August | 8.7 | 8.3 | 1.6 | 0.7 | 0.7 | 0.9 | 33.1 | 34.9 |
| September | 0.3 | 0.3 | -4.8 | 2.8 | 0.4 | 1.9 | 13.3 | 13.0 |
| October | 0.0 | 0.0 | -12.3 | 3.2 | -7.9 | 16.8 | 10.4 | 18.9 |
| November | 0.0 | 0.0 | -16.1 | 3.4 | -11.4 | 12.2 | 6.3 | 8.8 |
| December | 0.0 | 0.0 | -19.0 | 5.4 | -13.6 | 7.4 | 1.7 | 6.8 |

1  **Table 3. Mean and standard deviations of fluxes measured from the IMB data in the Beaufort Sea region,**

2  **in Wm$^{-2}$ in each month of the year. For each flux, the convention is that downwards=positive.**

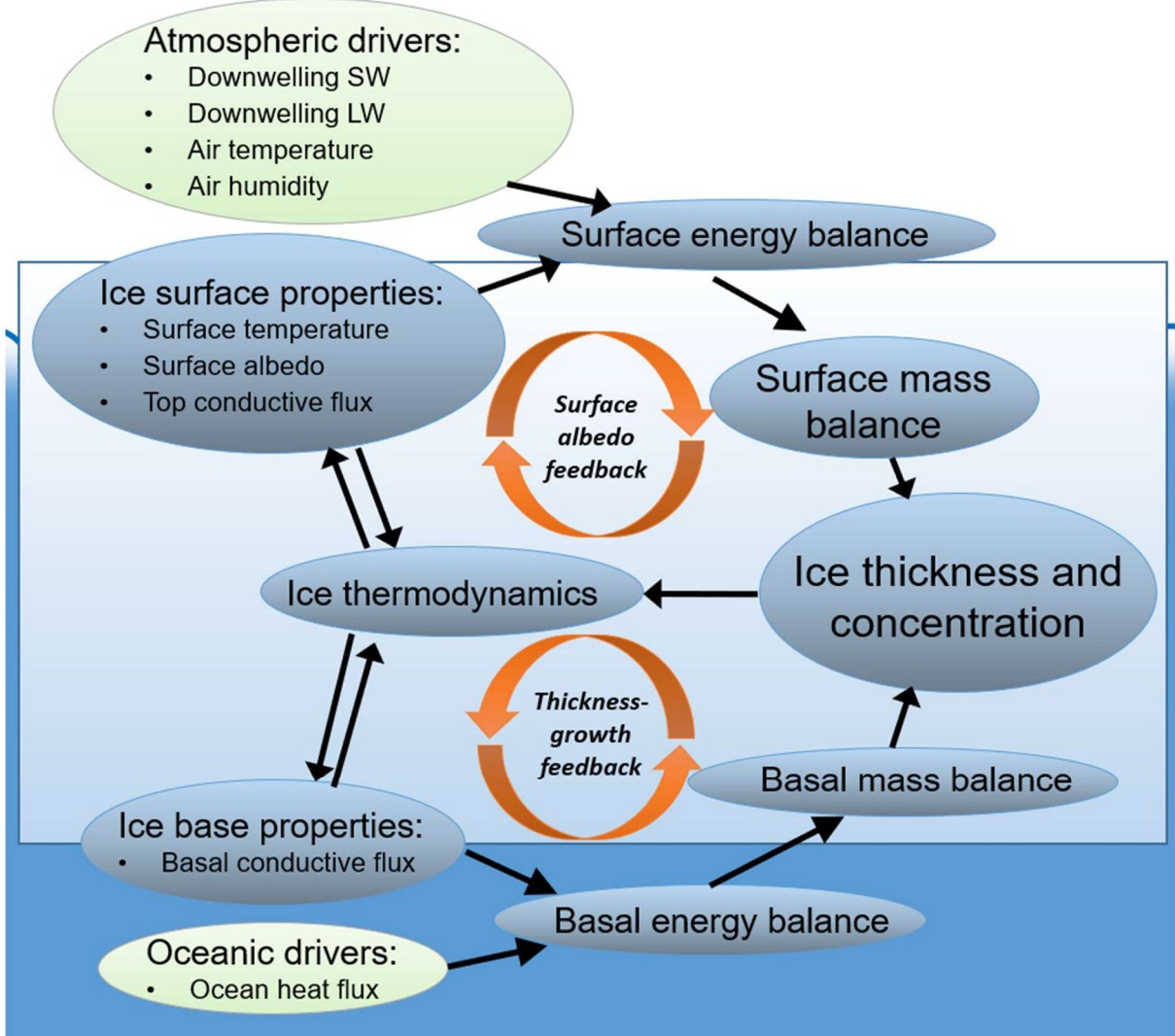

**Figure 1. Schematic demonstrating the causal links between ice thickness and extent, and energy and**

**mass balance at the top and basal surfaces of the sea ice.**

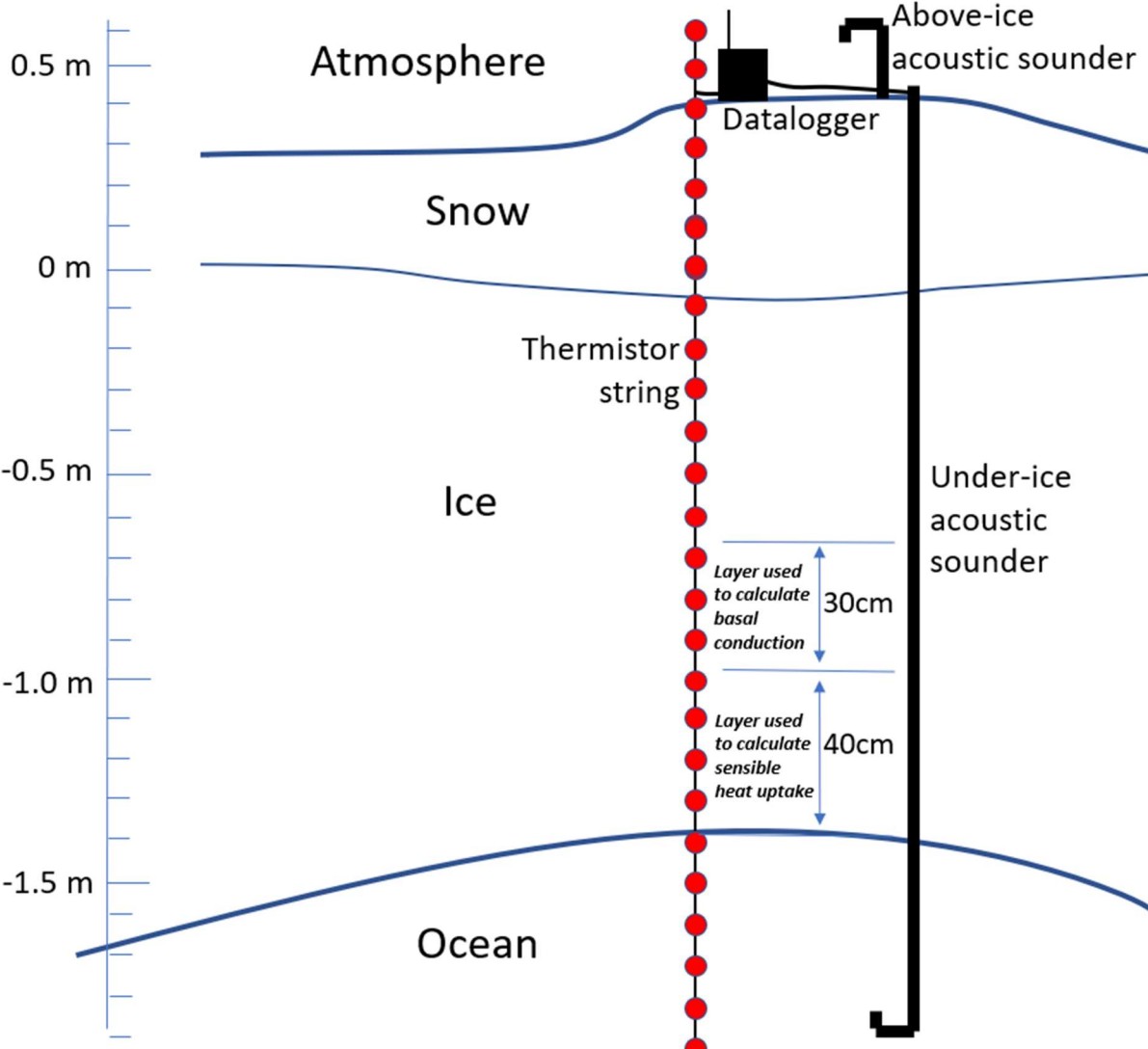

Figure 2. Diagram of the main components of an IMB, with layers used in this study for calculation of
fluxes at the base of the ice indicated. Adapted from Figure 1a of Planck et al (2019).

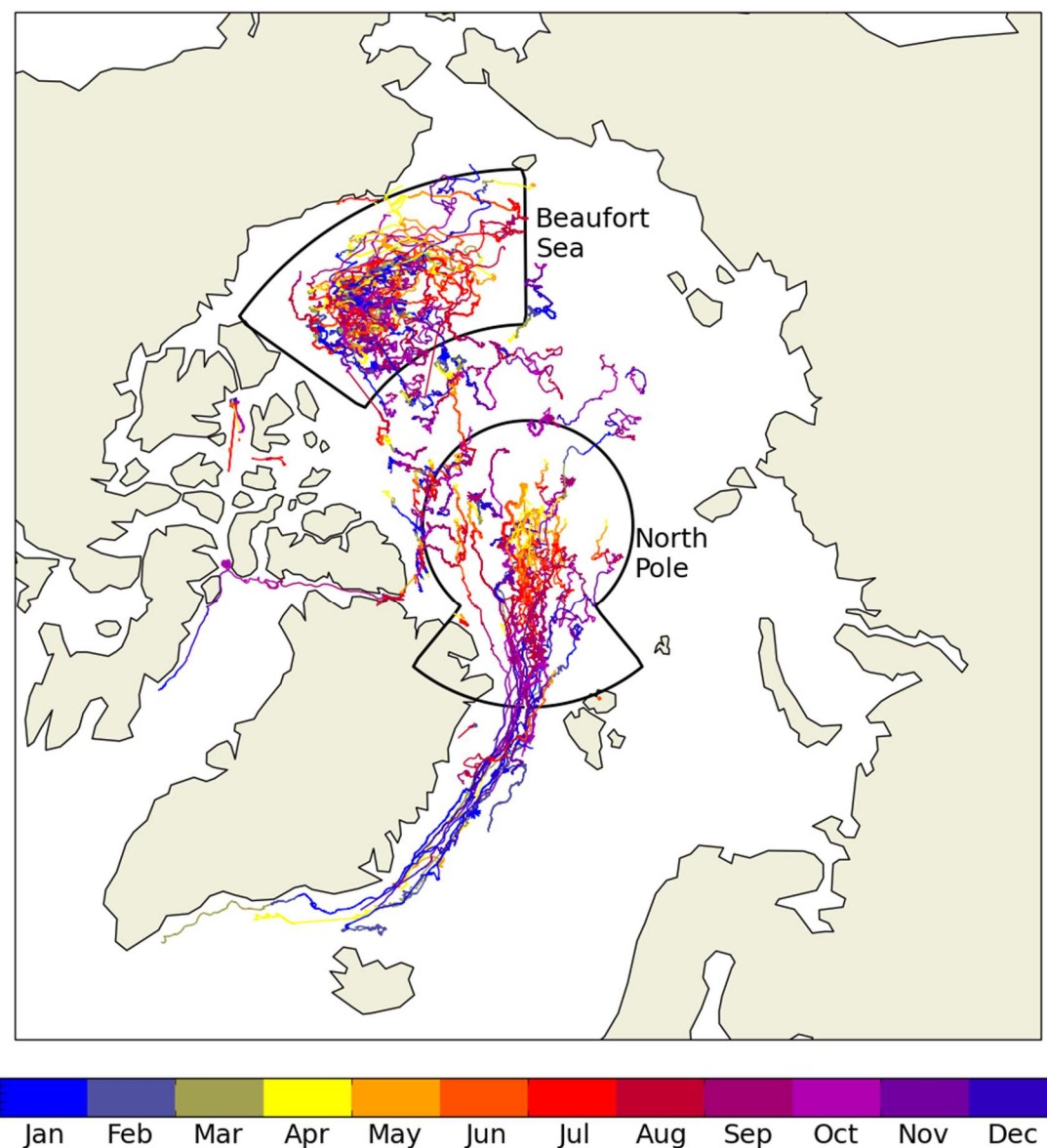

**Figure 3. The tracks of Arctic ice mass balance buoys from 1993-2015, with months of coverage indicated by the coloured shading. The North Pole and Beaufort Sea regions used in the analysis are shown by the thin black lines.**

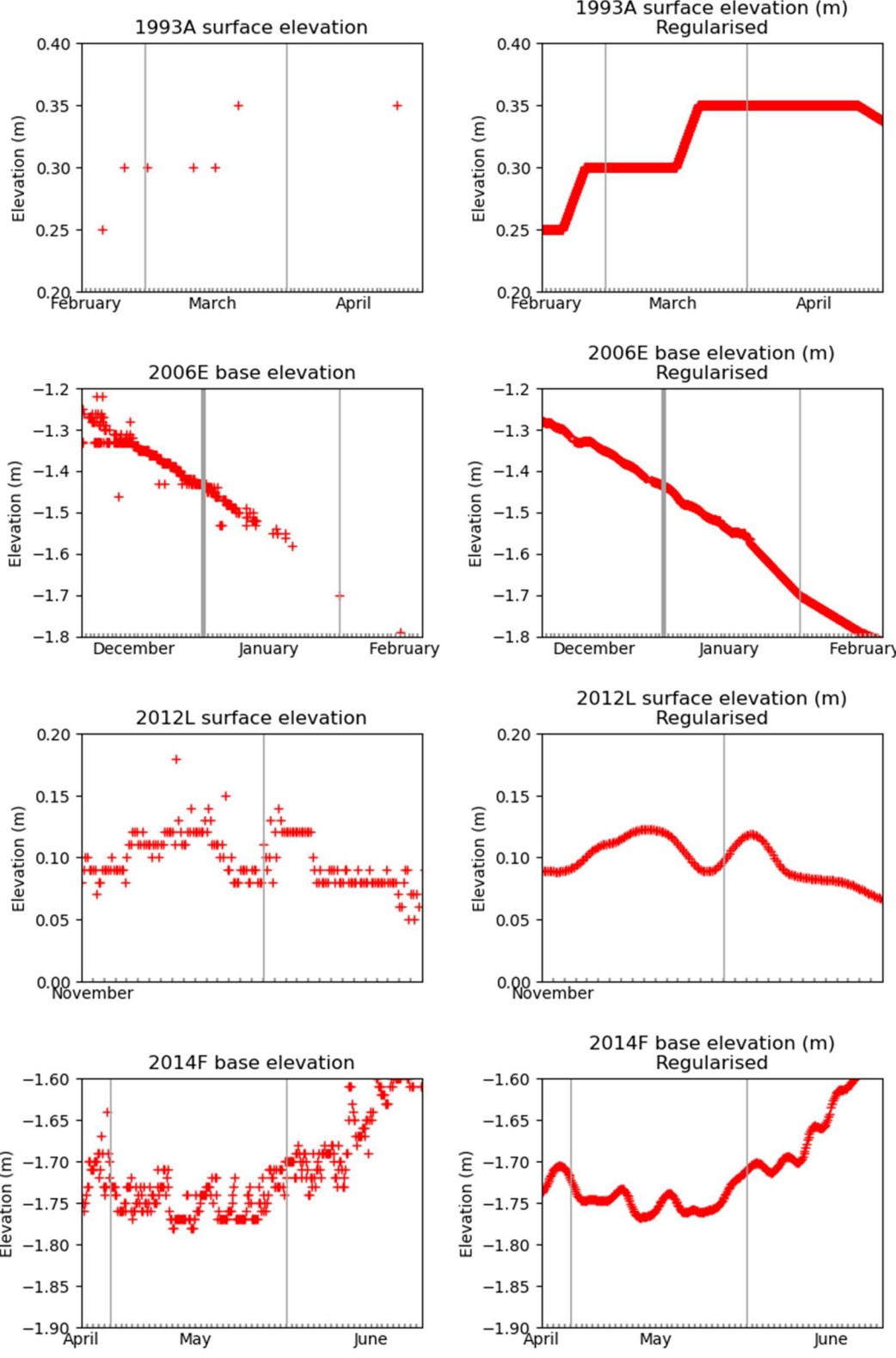

**Figure 4. Illustration of the regularisation process using four selected IMB data series. (Left) raw data;**
**(right) time series regularised to temperature measurement points.**

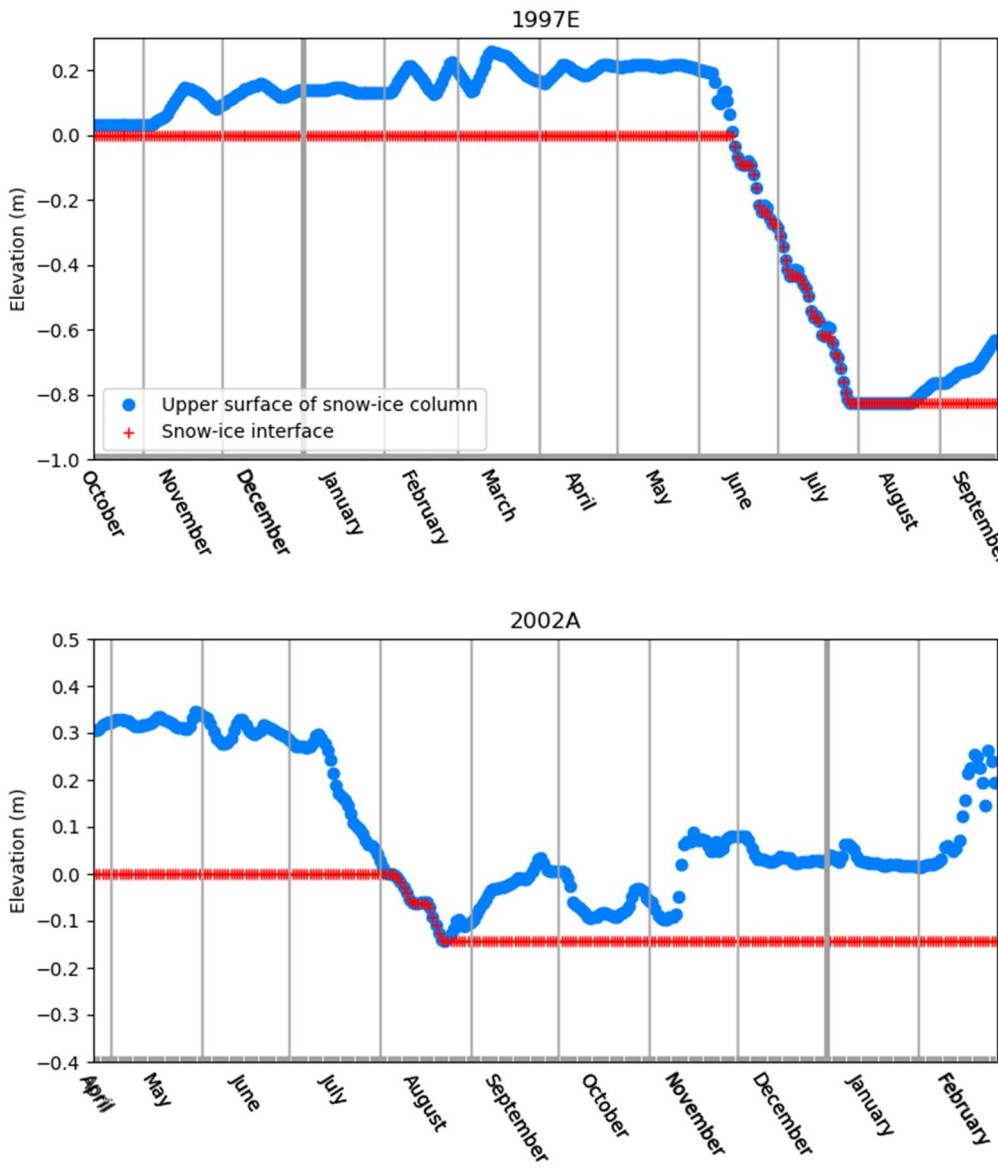

Figure 5. Two examples of estimating snow-ice interface from a regularised snow surface data series. The interface remains at a constant level unless the surface falls below this level, in which case the interface falls with the surface.

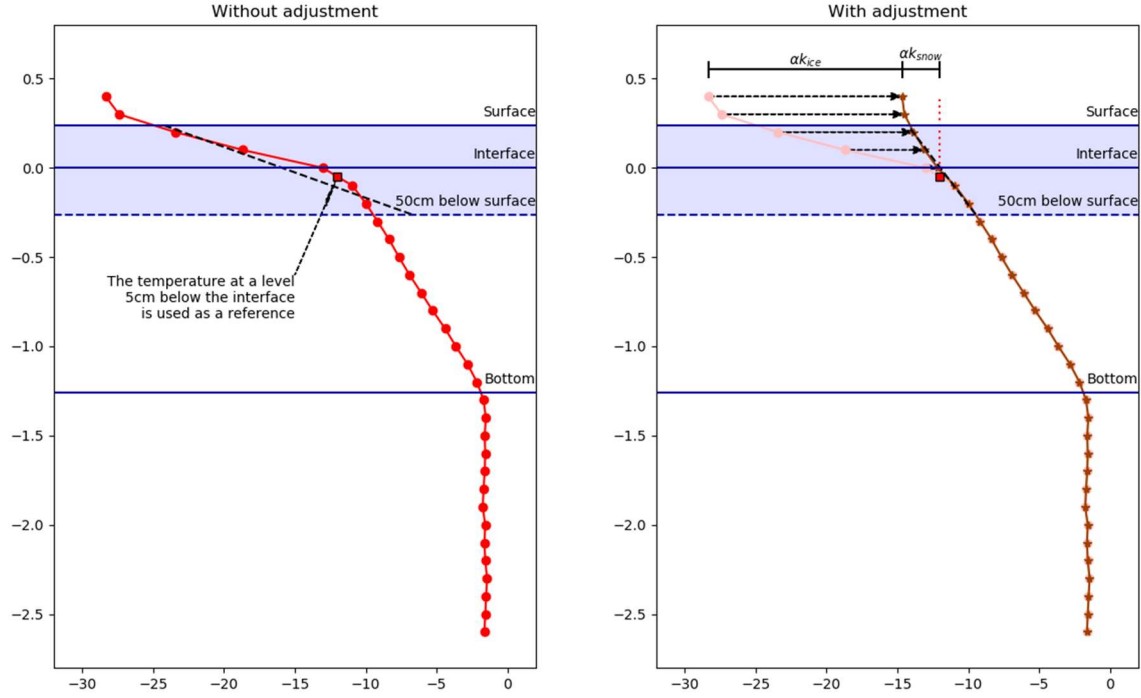

**Figure 6. Illustrating the process of estimating conductive flux across the top 50cm of the snow-ice column,**
**in the case that the snow-ice interface lies within this layer. The left panel shows the raw temperature**
**profile: taking a linear fit through these points does not produce a meaningful result because of the sharp**
**'corner' associated with the change in medium. The right panel shows the adjusted temperature profile:**
**the temperatures that would be expected if the snow layer were ice, temperature below the interface and**
**conductive fluxes remaining the same. The adjusted profile eliminates the corner, and a linear fit can be**
**taken. In the right panel, $k_{ice}$ denotes ice conductivity, $k_{snow}$ snow conductivity, and $\alpha$ an arbitrary**
**constant of proportionality.**

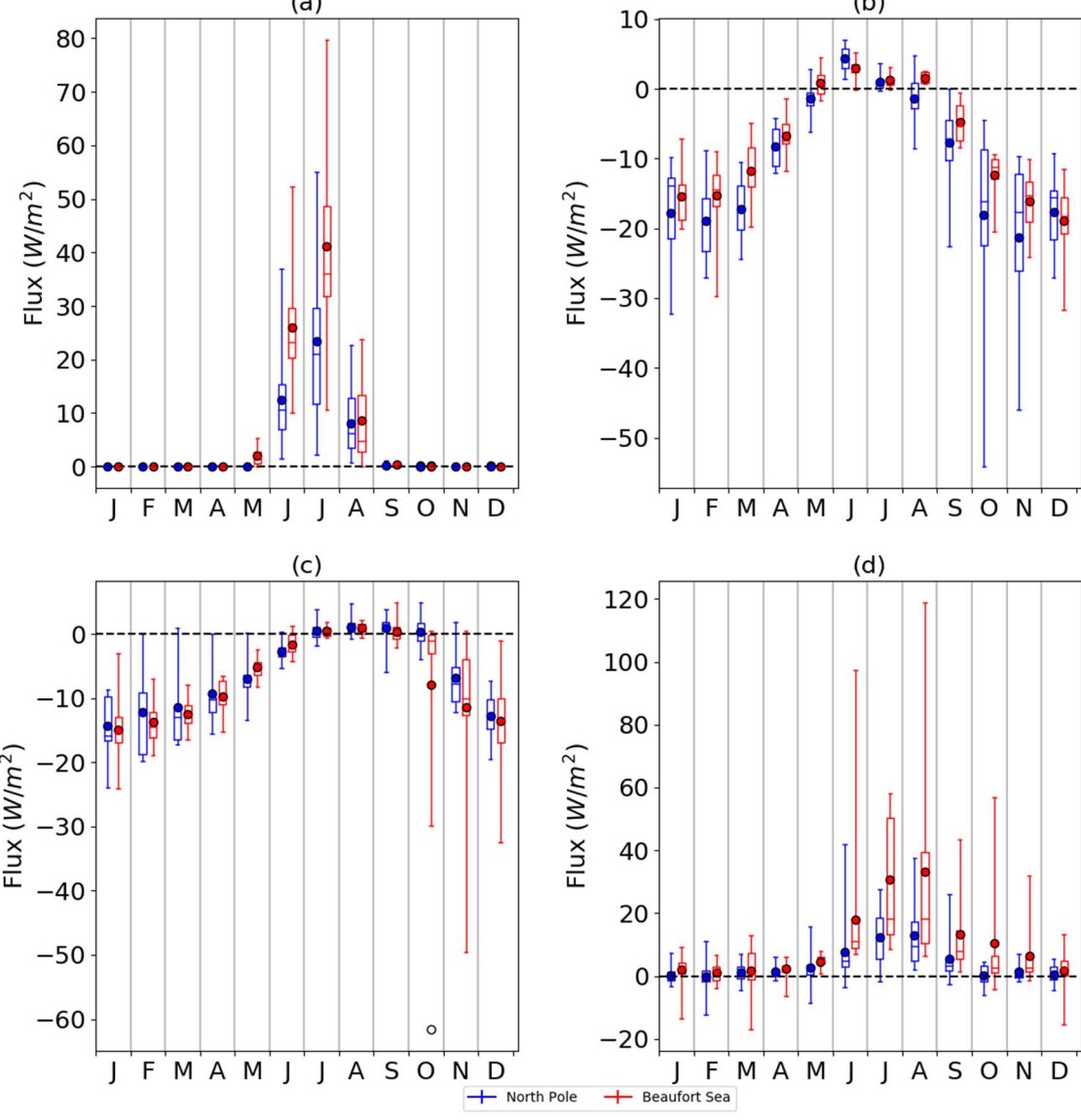

**Figure 7. Fluxes of (a) top melting, (b) top conductive flux, (c) basal conductive flux and (d) ocean heat flux,**
**estimated from the IMB data, shown for North Pole (blue) and Beaufort Sea (red) regions. For each month,**
**flux and region, the distribution is indicated by a boxplot showing range, interquartile range, median**
**(horizontal lines) and mean (filled circles). For all fluxes, the convention is that downwards=positive.**

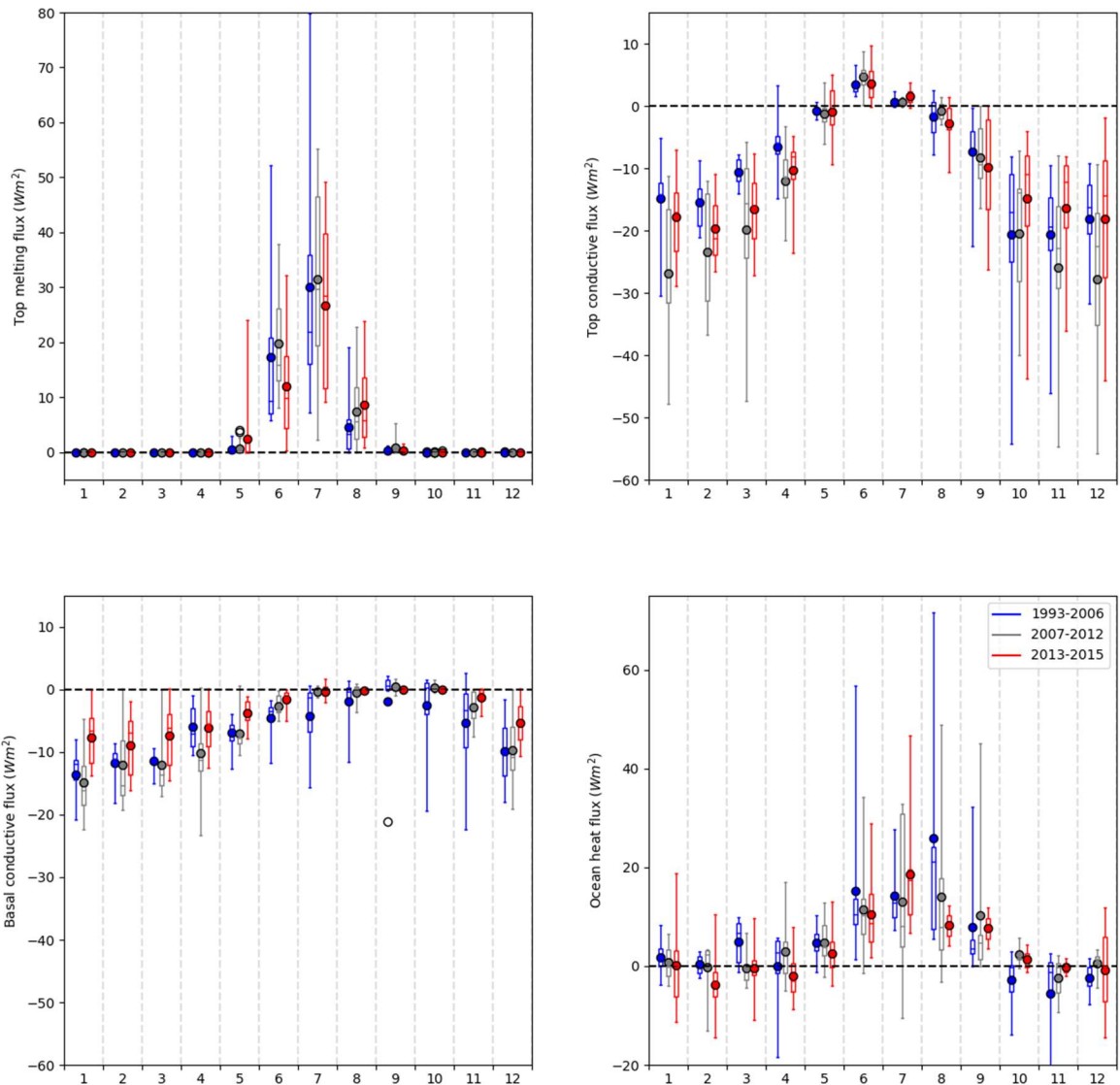

**Figure 8. IMB-measured distributions of (a) top melting; (b) top conductive flux; (c) basal conductive flux and (d) ocean heat flux, divided into the three periods 1993-2006, 2007-2012 and 2013-2015. Each distribution is illustrated with a boxplot showing range, interquartile range, median (horizontal lines) and mean (filled circles).**

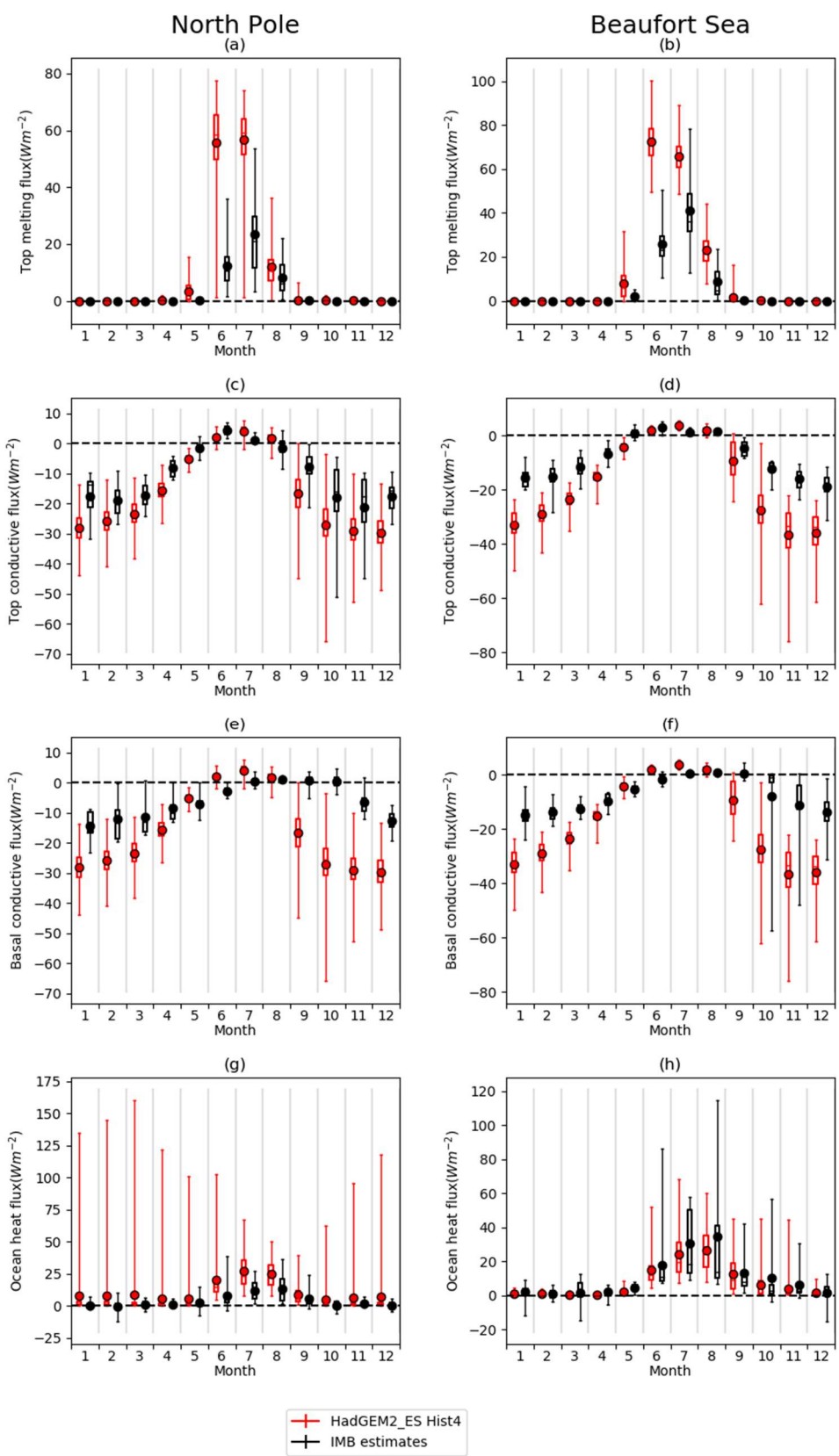

Figure 9. Comparing distribution of (a,b) top melt, (c,d) top conductive flux, (e,f) basal conductive flux and (g,h) ocean heat flux from HadGEM2-ES (red) to those estimated from the IMB data (black), for the North Pole (left) and Beaufort Sea (right) regions.

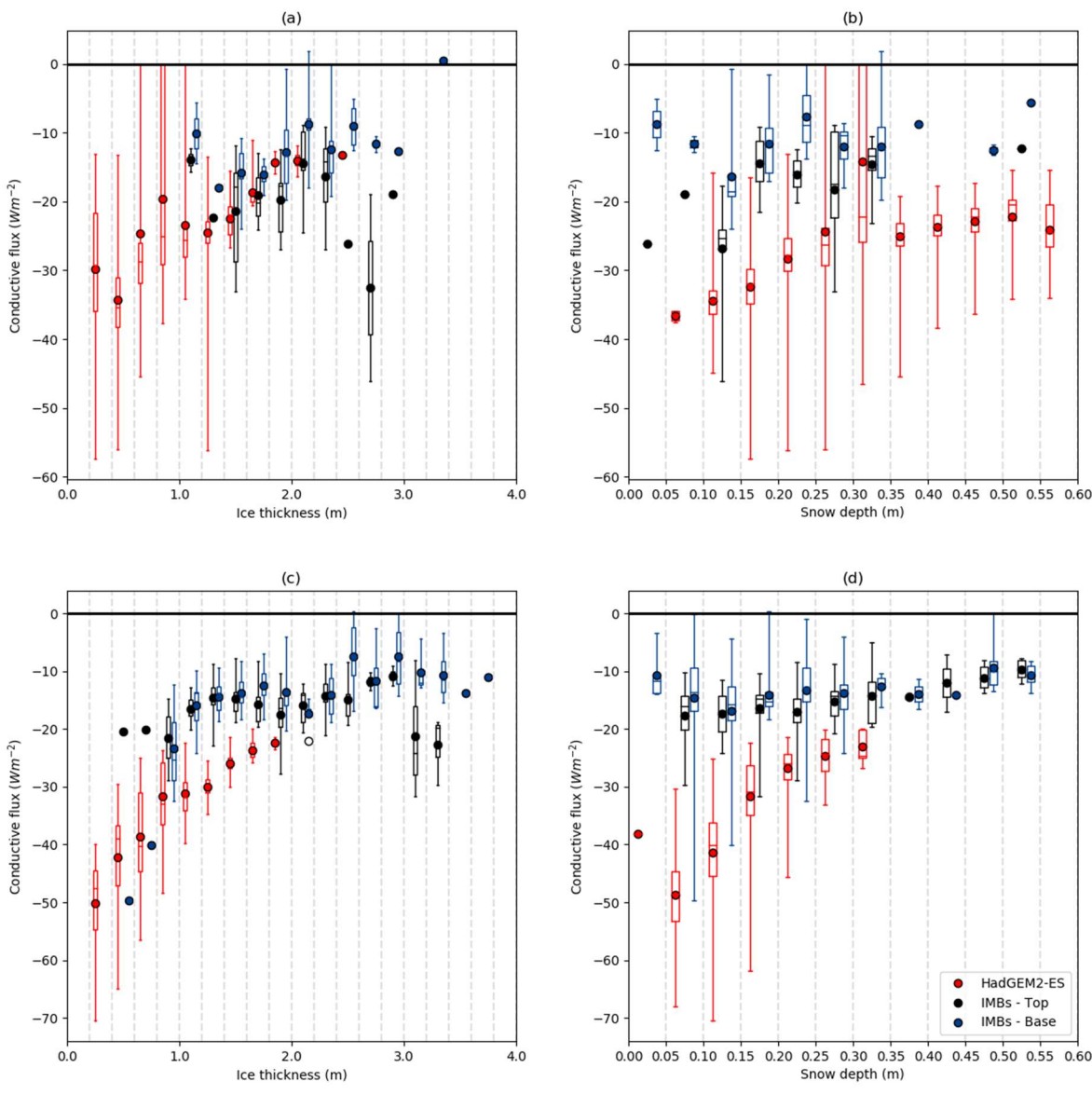

**Figure 10. Conductive fluxes plotted according to ice thickness (left column) and snow depth (right**
**column) for the North Pole (top) and Beaufort Sea (bottom) regions.**
**Figure S1. Measurements from the buoy 2015A, with a likely 'false bottom' being measured from early**
**April to late May, characterised by sudden step changes in estimated base elevation.**
**Figure S2. Ice fraction in the 5 thickness categories defined in Appendix A, estimated from submarine data**
**compiled by Rothrock et al. (2008) for the North Pole (solid lines) and Beaufort Sea (broken lines) regions.**

