# Peer review of "Using Arctic ice mass balance buoys for evaluation of modelled ice energy fluxes"

_Geoscientific Model Development, 2019_

## Referee Comment (RC1) · Anonymous Referee #1 · 20 Sep 2019

General comments:

The authors present a thorough analysis of the well known dataset of ice mass buoys deployed covering the Central Arctic and Beaufort Gyre regions since 1993 to provide climatology seasonal estimates of the top and bottom ice conductive and melt and ocean fluxes over and under sea ice. The novelty of the method lies in the fine analysis of the data and in the physical processing applied to retrieve meaningful fluxes that can then be used to evaluate climate models such as the HadGEM2-ES Met Office model. I am supportive of this paper being accepted in this general as this dataset and methodology offers a useful tool to the modelling but also remote sensing and in-situ communities. Nevertheless one of the main strength of this paper is the (clean) dataset produced as well as the algorithm developed to analyse these dataset and I strongly

encourage the authors to make these data available to the community. In addition, in this case, the nature of the product calls for more transparency and sharing the code and data will warrant easier reproducibility for the scientific community. Finally, a significant effort is needed to clarify the sensitivity of the analysis to the various constants and approximations made. I provide some detailed comments below on how this could be achieved.

Specific comments:

Abstract

1 Introduction

P1L23: add reference to Kwok, 2018

2 Calculating monthly-mean energy fluxes from the IMBs

P3L26: would a more advanced optimal interpolation scheme improve the results? P3L33: define z_srf and explain a little more (maybe in appendix or with figure 4 how zsrf and zint are sufficient to estimate both changes of surface and bottom sea ice. P4L2: King et al, 2018 and Mercouriadi et al, 2018 have shown that such snow ice formation is prevalent in some regions of the Arctic. Discuss. P4L13: couldn't you ask the data providers? P4L19: a link to the code would be very valuable here. P4L26: you can cite Alexandrov et al, 2010 for values of snow and ice density. Snow evolves throughout the season with values typically from ∼200 to 350 (i.e. Tilling et al, 2017) P4L34: not clear where this formula comes from and if it applies to the real snow on sea ice. At what depth? P5L9: this fixed thickness (say L) is a parameter of your analysis. Discuss how sensitive your results are to this choice. P5L29: similarly how does the uncertainty on these constants impact your results? Discuss. P6L1: These values come from where? Recent work Nandan et al 2018 show salinity at the snow ice interface to be larger than 1. There are more references buy Turner et al 2015 (model work on CICE) but also Notz etc. . . P6L5: equation is no readable P6L17-21:

where is that shown. Perform proper sensitivity analysis to all these parameters in your plots, discussion etc... P7L1: interesting. How would you inform the S value. Is it measured? Explain. What problem are you referring to here. P7L5: Tsamados et al, 2015 has implemented the three equation boundary conditions and discussed false bottom impact on sea ice - ocean bottom fluxes P7L13: Interesting. Can you see synoptic signal related to snow forcing (i.e. storms?). At what timescale are you resolving these? Monthly? Should you pre-process such erroneous signals before monthly averaging? Explain -> share code!

P3L28: Some have argued that power law is in the forcing? How sensitive are your results to the spatio-temporal lenghtscales of the atmo/ocean forcing? P4L8: we branch -> meaning? P4L13: justify this choice. Cite Landy et al, 2019 P4L19: bold not a good idea. i suggest run_ITD run_noITD

3 Deriving monthly-mean flux distributions from the IMBs

P7L27: why not two regions in the table 1 P7L28: why don't you discuss changes between decades I.e. 90s vs 00s vs 10s? P9L1: explain a bit more how these errors on the individual monthly scatter points are obtained. Are you performing an error propagation or are these simply a standard deviation?

4 Evaluatating modelled sea ice using the. IMB-derived fluxes

P9L21: not clear if you estimate the fluxes at the same location in time and space as the IMBs or average over the whole region for the whole month. You should both to test impact of IMB sampling on your results. P9L27: why didn't you perform your analysis on a more advanced model with more Ice thickness categories? P9L35: is it West 2018 or 2019. P10L4: again not clear if you perform comparison like for line (i.e. for same days and grid cells) or not. P10: here you list various fluxes but don't explain why you find these results. A bit too descriptive. P11L5: West 2018 or 2019? P11: discuss role of melt ponds (summer) and snow cover (winter) P12: I think a lot of your analysis is missing the link to melt pond coverage.

---

## Referee Comment (RC2) · Anonymous Referee #2 · 17 Oct 2019

**West et al.: Using Arctic ice mass balance buoys for evaluation of modelled ice energy fluxes**

The authors use ice mass balance buoys (IMB) to estimate fluxes through the top and bottom of sea ice. The authors present this new method and then compare the observed fluxes in the North Pole and Beaufort Sea regions. The authors then compare the observed fluxes with modeled fluxes from the HadGEM2-ES climate model. The main findings are that there are biases in these fluxes in the model, which are likely due to the biased mean state of the model. Additionally, there are differences in the fluxes in the Beaufort Sea and North Pole regions. I have a few major and moderate concerns about the way the model and observations are compared and these need to be addressed before I can recommend publication.

**Major concerns:**

1) Internal Climate Variability

The elephant in the room for a comparison between a climate model and observational data is the issue of internal climate variability (see references below), which you never mention, and leads me to have major concerns with your method. This is also relevant for when the Arctic will become ice free, which you mention in the introduction. Since HadGEM2-ES is a fully-coupled, freely-evolving climate model a single model experiment should not be expected to match the observed sea ice conditions. You do not mention using ensembles and where/how the observations fit in an ensemble spread.

Indeed, it appears you are comparing the mean climate model state with the mean observations (Fig. 8). This is not a particularly useful comparison – we know the model is biased from your previous work and therefore we expect to see biases in these mean fluxes as a result! Instead, a more useful analysis would be to evaluate situations when the model does have similar thicknesses to those observed, do the fluxes match the observations? That would tell us more about the processes going on in the model and how well they compare to those observed. You could do this by plotting the distribution of the conductive fluxes by thickness for the model and observations for a particular month or throughout the year.

Two additional comments on the model: a) It would be useful to quantify other relevant model biases to the sea ice mass budget like SST or ocean heat transport; b) You compare different years from the model (1980-1999) and observations (1997-2016). I know you did analysis about how the periods are different (Pg.12, lines 13-24) but why not just use the same years that presumably have comparable radiative forcing?

Kay et al. 2011 (doi: 10.1029/2011GL048008)

Kay et al. 2015 (doi: 10.1175/BAMS-D-13-00255.1)

Jahn et al. 2016 (doi: 10.1002/2016GL070067)

England et al. 2019 (10.1175/JCLI-D-18-0864.1)

2) Additional sources of uncertainty

You mention uncertainty in salinity as one of the big uncertainties in the IMB flux calculations. I think you need to mention that there are also large ranges in the observed snow and ice densities (see refs below) that could cause uncertainty in the retrievals. The values you use are reasonable, but you need to at least acknowledge this and do some basic calculations about how big a difference these values make.

Franz et al. 2019 (doi: 10.5194/tc-13-775-2019)

Webster et al. 2018 (doi: 10.1038/s41558-018-0286-7)

3) Significance

You spend much of section 3 describing differences in observations in two regions and sections 4.1/4.2 describing differences in the mean state of the model and observations. No significance tests were discussed to indicate whether the means are really significantly different in Figs. 7/8. A simple t-test should suffice, but without this I have a hard time believing some of the conclusions (e.g. that September fluxes differ in the IMB between regions).

**Moderate concerns:**

1) Model thermodynamics

I am surprised that HadGEM2-ES, a CMIP class climate model, uses the very simple zero-layer thermodynamics and I think that this should be addressed. The Bitz and Lipscomb thermodynamics is more realistic than zero-layer, and even this has been superseded by the mushy layer thermodynamics of Turner and Hunke (see below). I realize you can't change the model at this point and this shouldn't prevent publication, but your own results show that the assumptions of the zero-layer scheme for conductive fluxes are bad (Fig. 7 and Table 1). Maybe one of the conclusions should be that the zero-layer cannot represent the observed processes so HadGEM2-ES might want to stop using it?

Bitz and Lipscomb 1999 (doi: 10.1029/1999JC900100)

Turner and Hunke 2015 (doi: 10.1002/2014JC010358)

2) Figures
Individually these comments are fairly minor, but the sum of them is moderate.
- Table 1 – Please add the # values per month and per flux to this table rather than just listing them in the text. Also, adding the units below the flux names (not just in the table description) would be helpful. I expect top melt to usually be in cm/day (or kg m-2 s-1), not W/m2.
- Fig. 1 – It would be helpful to add the sign convention for fluxes here at the interfaces (show the flux direction for positive!). Either label or remove the red/yellow arrows.
- Fig.3 and Fig.4 – you don't have units on the y-axis or in the labels.
- Fig. 4 – Please define surface_r and interface_r. Is snow depth the difference between these two lines? Please clarify on the figure and in text.
- Fig. 5 – I think a diagram of an IMB would be helpful for modelers, which I think this one is, but it's poorly labeled. What do the dots represent (thermistors? Is this why they go above the snow interface)? What does the L/R position of the dots mean (temperature?)? Why is there a green line over part of the dots?

- Fig. 6 – the circle colors on this figure are very hard to make distinguish. Perhaps different shaped symbols in black would be better? Again, define surface_r, interface_r, and bottom_r. Are points where there is blue (aka surface_r) mean that the blue and green circles are overlapping? Adding arrows to indicate the transitions for the false bottom would also be helpful? This figure also makes me question why you don't linearly interpolate between the "correct" depth – it looks possible so why lose that data?
- Fig. 7 – It might be clearer to show the total spread in values with shading and then just a central dot for the mean since the individual points with their spread get hard to distinguish. Also, in text you list the means but they need to be shown to be significantly different.
- Fig. 8 – What purpose does this figure provide other than the model bias, which we already expected from your previous work. Again, if you do show it, significance tests are important. I think a PDF of the flux by thickness would be more helpful to supplement this figure.

3) Code availability

The effort put in here by the authors to make the data available is lackluster. I understand the model code itself may not be available. However, the authors should make more effort to list how to get the buoy data (raw and processed) as well as the model data (if the code isn't available) since these should be public if it's part of the CMIP archive. See the guidance on the website: https://www.geoscientific-model-development.net/for_authors/code_and_data_policy.html

**Minor concerns:**

- Shu et al. 2015 isn't in your references.
- Pg.4, line 6-8 – do you mean the ice temperature or air temperature? What's going on to cause the non-physical temperatures?
- Pg.4 line 14 - How long do buoys last? Give a range here.
- On page 7 you state the convention of positive fluxes indicate downwards. It would be very helpful if you mention that much earlier on Page 4 line 21. And put this on a diagram (Fig.1 and/or Fig.5) too.
- Pg.7 line 20 – how thin is "quite thin"? Be specific!
- Pg.7 line 26-28 – These means are over all available buoys and all years, right? It would be good to be explicit.
- Pg.9 line 15 – change "7 to 143" W/m2 because negative fluxes are possible and the current wording is unclear.
- Pg.8 line 25 AND Pg.10 line 1, mention that those regions are defined in Fig.2.
- Pg.12, line 5 – what is the model grid (it hasn't been mentioned)? Why the huge range in grid cell size?

---

## Author Comment (AC1) · 18 Nov 2019

**Authors' response to reviews of 'Using Arctic ice mass balance buoys for evaluation of modelled 1 ice energy fluxes'**

We thank the reviewers for their detailed, considered analysis of our manuscript. Our response is in three sections:

1. Response to reviewer 1
2. Response to reviewer 2
3. Summary of proposed changes and additions to manuscript arising from the reviews

**1. Reply to reviewer 1**

Original reviewer comments are shown in italic font, our response in normal font.

*General comments: The authors present a thorough analysis of the well known dataset of ice mass buoys deployed covering the Central Arctic and Beaufort Gyre regions since 1993 to provide climatology seasonal estimates of the top and bottom ice conductive and melt and ocean fluxes over and under sea ice. The novelty of the method lies in the fine analysis of the data and in the physical processing applied to retrieve meaningful fluxes that can then be used to evaluate climate models such as the HadGEM2-ES Met Office model. I am supportive of this paper being accepted in this general as this dataset and methodology offers a useful tool to the modelling but also remote sensing and in-situ communities. Nevertheless one of the main strength of this paper is the (clean) dataset produced as well as the algorithm developed to analyse these dataset and I strongly encourage the authors to make these data available to the community. In addition,in this case, the nature of the product calls for more transparency and sharing the code and data will warrant easier reproducibility for the scientific community. Finally, a significant effort is needed to clarify the sensitivity of the analysis to the variousconstants and approximations made. I provide some detailed comments below on how this could be achieved.*

We thank the reviewer for their kind remarks. Although it was not possible to publish the code in time for the publication of the discussion paper, due to internal procedures, we strongly agree that the code should be published, and intend to publish a version with the next revision of the paper (providing one is requested), as long as the internal review process can be completed in time.

The reveiwer's suggestion that, in addition, a version of the processed data should be made available to the community is constructive and helpful. After reading this, we have begun an overhaul of the code in order to enable the production of the processed IMB data in netCDF format, and hope to be able to publish this simultaneously.

*Specific comments:*

*Abstract*

*Introduction*

*P1L23: add reference to Kwok, 2018*

This will be added.

*Calculating monthly-mean energy fluxes from the IMBs*

*P3L26: would a more advanced optimal interpolation scheme improve the results?*

Yes, probably – such a scheme could make use of information about other variables, for example. However we decided early on to use a very simple scheme for the present study to ease the processing of data. The results of the estimation scheme used were, individually, sensible. The estimation scheme is contained in a single function in the code, which could easily be replaced by a more advanced scheme in a future study.

*P3L33: define z_srf and explain a little more (maybe in appendix or with figure 4) how zsrf and zint are sufficient to estimate both changes of surface and bottom sea ice.*

The point to explain is how $z_{sfc}$ is sufficient to estimate changes in the snow-ice interface elevation (not in the surface – which $z_{sfc}$ represents – or in the ice base). However, clearly this point is not explained very clearly. $z_{sfc}$ will be defined and we will try to improve the clarity of this paragraph.

*P4L2: King et al, 2018 and Mercouriadi et al, 2018 have shown that such snow ice formation is prevalent in some regions of the Arctic. Discuss.*

We note that tracks of floes analysed for the N-ICE2015 expedition, on which significant snow-ice formation is observed (as described by the references you cite), cross the far south-east of our North Pole region, in several cases intersecting IMB tracks. This apparent contradiction needs resolving.

Our explanation of why we do not believe snow-ice formation is prevalent in the IMB dataset requires further elaboration. For most months of the IMB dataset, the ice is sufficiently thick, and the snow sufficiently thin, that snow-ice formation can be judged unlikely from hydrostatic principles. In addition, during the winter, at randomly inspected times and buoys when the surface temperature is sufficiently cold, the change in temperature gradient associated with the change in conductivity between ice and snow is roughly co-located with the snow-ice interface (as measured by the IMBs, or estimated by our method). 'Roughly' meaning to within about 10cm.

There are isolated instances (4 by our count) where snow-ice formation is possible, but these are often associated with sensor failure. For example, the buoy 2015D sees steep rises in snow depth in November 2015 that would, if genuine, almost certainly be associated with snow-ice formation; but as all other elevation series begin reporting missing data at the same time, and many of the thermistors in the ice layer appear to stop working, no data from this buoy reaches the final dataset in any case. This issue could suggest that snow-ice formation had the potential to corrupt the IMB sensors, and that therefore it is significantly undersampled by the IMBs.

We will add a more detailed discussion of the snow-ice formation issue, along the lines of the paragraphs above.

*P4L13: couldn't you ask the data providers?*

This would be a sensible way to proceed; we have contacted them, and hope that this will enable a more definitive answer to this question.

*P4L19: a link to the code would be very valuable here.*

The code will be provided with the revised version of the document.

*P4L26: you can cite Alexandrov et al, 2010 for values of snow and ice density. Snow evolves throughout the season with values typically from ~200 to 350 (i.e. Tilling et al, 2017)*

We will cite the paper the reviewer suggests. Reviewer 2 suggested that it would be a useful exercise to judge the sensitivity of the fluxes to assumptions about snow density.

*P4L34: not clear where this formula comes from and if it applies to the real snow on sea ice. At what depth?*

This refers to another paragraph that was clearly not very well written; perhaps a new figure would help clarify the meaning here. Conductive fluxes are determined by taking linear fits through the temperature profile: near the snow-ice interface the temperature profile 'turns a corner', rendering a linear fit useless. The point of the formula is to 'straighten' the temperature profile by rotating the part of the profile above the interface. In other words, we calculate what the temperature of the snow layer would be if it was ice instead, and the conductive flux profile remained the same. Given this intent, the formula is just basic geometry: the issue as regards physical realism is with the underlying assumptions about conductivity, rather than the formula itself.

*P5L9: this fixed thickness (say L) is a parameter of youranalysis. Discuss how sensitive your results are to this choice.*

This would be a valuable addition to the analysis and we will include this in the revised version.

*P5L29: similarly how does the uncertainty on these constants impact your results? Discuss.*

We agree that the uncertainty analysis would be improved by consideration in uncertainty in the thermodynamic parameters of sea ice. However, our view is that not all of the parameters are equal in this respect. The ratio between ice salinity and melting temperature is a very well-constrained parameter and is unlikely to contribute to the uncertainty in any meaningful way. By contrast, the rate of dependence of conductivity on salinity (denoted by {\beta} in our paper) is subject to considerable uncertainty which would bear investigation. For example, it would be interesting to see how our flux estimates vary when the alternative formulation of Pringle et al (2009) is used.

The salinity, heat capacity and specific heat of fusion of fresh ice are also constants that would bear investigation in this regard. However, we think that it is likely that uncertainty in salinity would still dominate the uncertainty in the IMB-derived fluxes, particularly if we use expanded salinity ranges, as discussed below.

*P6L1: These values come from where? Recent work Nandan et al 2018 show salinity at the snow ice interface to be larger than 1. There are more references buy Turner et al 2015 (model work on CICE) but also Notz etc…*

We thank the reviewer for providing these references. Our view is that the subject of sea ice salinity is sufficiently complex that the only way of properly accounting for this in the present study, without seriously detracting from its main purpose, is to use uncertainty ranges that encompass all realistic salinity values. We will use expanded uncertainty ranges in the next paper revision, allowing values up to 10 at both the top and lower surfaces of the ice.

*P6L5: equation is no readable*

This will be corrected.

*P6L17-21 where is that shown. Perform proper sensitivity analysis to all these parameters inyour plots, discussion etc…*

The reviewer has correctly identified that there are indeed additional sources of uncertainty not accounted for here, and we will endeavour to incorporate these into our estimates.

*P7L1: interesting. How would you inform the S value.Is it measured? Explain. What problem are you referring to here.*

Salinity is not measured, but the temperature and elevation data act to constrain the salinity ranges. For example, if the ice surface is at -0.1m, and the temperature at -0.2m is -0.1 deg C, this implies the melting temperature of the ice at -0.2m is greater than -0.1 deg C. Hence the salinity is lower than 1.9. The 'problem' as described in this paragraph is that occasionally the temperatures are in this way inconsistent with the assumed salinity ranges.

*P7L5: Tsamados et al, 2015 has implemented the three equation boundary conditions and discussed false bottom impact on sea ice - ocean bottom fluxes*

We will add a reference to their analysis. Instead of our stating that false bottom formation renders the computation of ocean heat flux impossible, it is probably more accurate to say that it greatly complicates its calculation – and given the number of data points affected, may be outside the scope of this study.

*P7L13: Interesting. Can you see synoptic signal related to snow forcing (i.e. storms?). At what timescale are you solving these? Monthly? Should you pre-process such erroneous signals before monthly averaging? Explain -> share code!*

At the moment we are processing these at monthly timescale. We agree that it would be preferable to process these before monthly averaging, and will try to improve on this in our next revision as part of our code overhaul.

*P3L28: Some have argued that power law is in the forcing? How sensitive are your re-sults to the spatio-temporal lenghtscales of the atmo/ocean forcing? P4L8: we branch-> meaning? P4L13: justify this choice. Cite Landy et al, 2019 P4L19: bold not a goodidea. i suggest run_ITD run_noITD3*

We assume that this paragraph is included in error as it does not obviously correspond to text in the paper (and the chronology is different).

**Deriving monthly-mean flux distributions from the IMBs**

*P7L27: why not two regions in the table*

We did not want to include too much information in a single table, to improve ease of reading. Additional tables could be provided, giving the flux distributions by region.

*P7L28: why don't you discuss changes between decades I.e. 90s vs 00s vs 10s?*

Only 7 buoys are available for the 1990s; 6 of these were from the SHEBA campaign (i.e. in the same year, 1997-1998, at the same location in the Beaufort Sea), and the remaining buoy, deployed in 1993, was also located in the Beaufort Sea. Hence there is probably not enough data from this decade to properly sample spatial or interannual variability in the Beaufort Sea region, and none at all in the North Pole region.

After the year 2000, and particularly after the year 2007, buoy numbers increase substantially. There may be scope to carry out some limited analysis of interannual variability by comparing, for example, the period 1993-2006 to 2007-2015, particularly if data from different months and regions can be aggregated. However the periods will need to be chosen with care to ensure adequate sampling. We will see if there is some way to do this in the revised manuscript.

*P9L1: explain a bit more how these errors on the individual monthly scatter points are obtained. Are you performing an error propagation or are these simply a standard deviation?*

The errors on the individual monthly scatter points are obtained by propagating the salinity uncertainties assumed in Section 2 (1 +/- 1 at ice top and 4 +/- 4 at ice base) and the measurement uncertainties described by Perovich and Richter-Menge (2006). These should not be

confused with the intervals given in e.g. page 9 line 1, which describes the standard deviation of the full distribution of central estimates of monthly mean fluxes.

**4 Evaluating modelled sea ice using the. IMB-derived fluxes**

*P9L21: not clear if you estimate the fluxes at the same location in time and space as the IMBs or average over the whole region for the whole month. You should both to test impact of IMB sampling on your results.*

We average over the whole region. Model internal variability is such that we do not expect the model to exactly capture the conditions at each point in space and time, and therefore did not see any particular value in sampling the model only at identical points to the IMBs. As discussed in the appendix, we suspect that the largest impact of IMB sampling is through the ice thickness – this would not be solved by evaluating the model at the same points in space and time, as we would still be evaluating fluxes over the entire grid cell. It is not clear to us, therefore, that the impact of IMB sampling would be revealed by the methods that the reviewer suggests.

*P9L27: why didn't you perform your analysis on a more advanced model with more Ice thickness categories?*

We chose to evaluate HadGEM2-ES because its sea ice simulation was already fairly well-understood. Confidence in the IMB-based evaluation could therefore be informed by how consistent this was with the sea ice and surface radiation evaluation. We are now evaluating the new UK CMIP6 models (HadGEM3-GC3.1 and UKESM1.0) in the same way, but this evaluation appears to be outside the scope of this study, which is intended only to demonstrate the new method. Note that although the new models are more advanced (multilayer thermodynamics and explicit meltponds), the number of thickness categories is the same.

*P9L35: is it West2018 or 2019.*

2019. This will be corrected.

*P10L4: again not clear if you perform comparison like for line (i.e. forsame days and grid cells) or not.*

No – the model distribution is calculated over the whole region, for reasons described above. Model internal variability means that we do not expect fluxes at the exact same pathways, at the same times, to better represent the conditions than the fluxes over the whole region and time period.

*P10: here you list various fluxes but don't explain why you find these results. A bit too descriptive.*

The intention here was to separate the evaluation (section 4.1) with the discussion of reasons for the results (section 4.2). Some discussion could instead be added to section 4.1 if this helps readability.

*P11L5: West 2018 or 2019?*

2019 – again this will be corrected.

*P11: discuss role of melt ponds (summer) and snow cover (winter)*

The role of snow cover in winter in the conductive flux biases can be investigated directly by comparing IMB-measured snow depths to those modelled by HadGEM2-ES. Indeed, there may be scope to carry out such a comparison alongside a comparison of conductive flux and ice thickness which was requested by Reviewer 2 (see below).

The role of melt ponds in summer in the top melt biases (which is what we assume the reviewer means) would be more difficult to evaluate directly. In HadGEM2-ES, meltponds are parameterised from surface temperature (and can therefore be diagnosed from this variable). In the IMBs however, surface temperature is probably a highly imperfect proxy for meltpond coverage (and hence surface albedo). In theory, meltpond occurrence could be detected

*P12: I think a lot of your analysis is missing the link to melt pond coverage*

**2. Reply to reviewer 2**

*The authors use ice mass balance buoys (IMB) to estimate fluxes through the top and bottom of sea ice. The authors present this new method and then compare the observed fluxes in the North Pole and Beaufort Sea regions. The authors then compare the observed fluxes with modeled fluxes from the HadGEM2-ES climate model. The main findings are that there are biases in these fluxes in the model, which are likely due to the biased mean state of the model. Additionally, there are differences in the fluxes in the Beaufort Sea and North Pole regions. I havea fewm ajorand moderate concerns about the way the model and observations are compared and these need to be addressed before I can recommend publication.*

**Major concerns**

*1) Internal Climate Variability*
*The elephant in the room for a comparison between a climate model and observational data is the issue of internal climate variability (see references below), which you never mention, and leads me to have major concerns with your method. This is also relevant for when the Arctic will become ice free, which you mention in the introduction. Since HadGEM2-ES is a fully-coupled, freely-evolving climate model a single model experiment should not be expected to match the observed sea ice conditions. You do not mention using ensembles and where/how the observations fit in an ensemble spread.*

In the analysis shown in this paper, we use only the first historical ensemble member of HadGEM2-ES. We can revise the analysis to include the full ensemble (albeit only 4 members) to estimate the internal variability in the fluxes, and compare these to the model biases.

However, it's important to note that the main purpose of the study is to demonstrate the value of the IMB-based evaluation, rather than to draw conclusions from the model biases demonstrated. To demonstrate the value of the evaluation method, it is in our view sufficient that a) the IMB-measured model biases are consistent with biases in the sea ice and surface radiation simulations, and b) they are larger in many months than the observational uncertainty in the IMB fluxes. Whether or not the biases result from internal variability, or another cause, is in our view of only secondary importance for this study. This would not be the case if we were trying to draw conclusions from the model biases, in which case the internal variability context would be vital (hence the many studies of internal variability in sea ice extent trends, some of which the reviewer quotes).

*Indeed, it appears you are comparing the mean climate model state with the mean observations (Fig. 8). This is not a particularly useful comparison –we know the model is biased from your previous work and therefore we expect to see biases in these mean fluxes as a result!*

But we would argue that to find biases in the mean fluxes that are physically consistent with biases in the sea ice simulation is in itself a significant result – because we are demonstrating an entirely new method of model evaluation. We will state this more explicitly.

*Instead, a more useful analysis would be to evaluate situations when the model does have similar thicknesses to those observed, do the fluxes match the observations? That would tell us more about the processes going on in the model and how well they compare to those observed. You could do this by plotting the distribution of the conductive fluxes by thickness for the model and observations for a particular monthor throughout the year.*

A comparison of conductive fluxes to ice thickness, for both the model and IMBs, would be an informative addition to the paper and we will incorporate this into a revision, possibly in a new subsection to section 4. However, there is an issue to be aware of.

A scatter plot of monthly mean IMB conductive fluxes and ice thicknesses against monthly gridbox mean conductive fluxes and ice thickness is still not a like-for-like comparison. This is mainly due

to the sea ice thickness distribution of HadGEM2-ES. A cell with mean thickness-over-ice of 3m, for example, could include 10% ice at 0.5m thick, 60% ice at 3m and 30% ice at 3.9m. This matters, of course, because the relationship between ice thickness and conductive flux is nonlinear. Average conductive flux in such a grid cell would be much higher than that in a single IMB-observed ice column of thickness 3m under identical atmospheric conditions, because the small amount of ice of thickness 0.5m would transfer much more energy to the atmosphere.

To solve this problem, and obtain a like-for-like comparison, the modelled flux needs to be plotted against the area-weighted harmonic mean thickness over ice, rather than the arithmetic mean. Because of the need to explain the issue, and justify the solution, the new subsection will not be brief.

*Two additional comments on the model: a) It would be useful to quantify other relevant model biases to the sea ice mass budget like SST or ocean heat transport.*

Ocean heat transport can be quantified. Over the Arctic Ocean as a whole it is about 4 Wm-2, roughly consistent with observational estimates to first order. We can also quantify it seasonally over the North Pole and Beaufort Sea regions to add context to the model biases. As above however, we would argue that this is only tangential to the purpose of the paper, which is to demonstrate the IMB-based evaluation. OHT is more relevant when discussing the causes of the model biases. But it is probably useful context so we will give it.

*b) You compare different years from the model (1980-1999) and observations (1997-2016). I know you did analysis about how the periods are different (Pg.12, lines 13-24) but why not just use the same years that presumably have comparable radiative forcing?*

We chose the period of 1980-1999 for consistency with the earlier study of HadGEM2-ES sea ice and surface radiation. The historical ensembles of HadGEM2-ES actually end in 2005, so it would be necessary to use a scenario experiment to get a comparable time period.

*2) Additional sources of uncertainty*

*You mention uncertainty in salinity as one of the big uncertainties in the IMB flux calculations. I think you need to mention that there are also large ranges in the observed snow and ice densities (see refs below) that could cause uncertainty in the retrievals. The values you use are reasonable, but you need to at least acknowledge this and do some basic calculations about how big a difference these values make. Franz et al. 2019 (doi: 10.5194/tc-13-775-2019) Webster et al. 2018 (doi: 10.1038/s41558-018-0286-7)*

This is a good idea and we will add a density component to the uncertainties in any subsequent paper revision.

*3) Significance*

*You spend much of section 3 describing differences in observations in two regions and sections 4.1/4.2 describing differences in the mean state of the model and observations. No significance tests were discussed to indicate whether themeans are really significantly different in Figs. 7/8. A simple t-test should suffice, but without this I have a hard time believing some of the conclusions (e.g. that September fluxes differ in the IMB between regions).*

This would also be a valuable addition. A Welch t-test would appear to be appropriate, as in most cases the sample sizes will not be the same and the variances of the distributions cannot be assumed to be equal.

**Moderate concerns**

*1) Model thermodynamics*

*I am surprised that HadGEM2-ES, a CMIP class climate model, uses the very simple zero-layer thermodynamics and I think that this should be addressed. The Bitz and Lipscomb thermodynamics is more realistic than zero-layer, and even this has been superseded by the mushy layer thermodynamics of Turner and Hunke (see below). I realize you can't change the model at this point and this shouldn't prevent publication, but your own results show that the assumptions of the zero-layer scheme for conductive fluxes are bad (Fig. 7 and Table 1). Maybe one of the conclusions should be that the zero-layer cannot represent the observed processes so HadGEM2-ES might want to stop using it?*

*Bitz and Lipscomb 1999 (doi: 10.1029/1999JC900100) Turner and Hunke 2015 (doi: 10.1002/2014JC010358)*

The new UK CMIP6 models, HadGEM3-GC3.1 and UKESM1.0, use the multilayer thermodynamics formulation of Bitz and Lipscomb (1999); a further paper is planned to compare these to HadGEM2-ES. But our view is that they are not relevant for the present study, which is intended to demonstrate a new method of evaluation rather than to compare models.

*2) Figures*

*Individually these comments are fairly minor, but the sum of them is moderate.*

*• Table 1 –Please add the # values per month and per flux to this table rather than just listing them in the text. Also, adding the units below the flux names (not just in the table description) would be helpful. I expect top melt to usually be in cm/day (or kg m-2s-1), not W/m2.*

Adding number of samples and units are both good suggestions to aid clarity, and will be carried out. However our view is that changing top melt units to cm / day would be unhelpful. This is because cm/day as a unit is less meaningful for the conductive and ocean heat fluxes, as these fluxes are not automatically converted to sea ice melt. And it's obviously desirable that the units should be consistent throughout the table.

*Fig. 1 – It would be helpful to add the sign convention for fluxes here at the interfaces (show the flux direction for positive!). Either label or remove the red/yellow arrows.*

The red and yellow arrows denote radiative fluxes, and will be labelled. We are not sure that a sign convention is necessary for a figure that is purely schematic, as the figure does not attempt to show any numbers. Possibly the sign convention for later figures (particularly 7 and 8) could be more clearly given, hosever.

*Fig.3 and Fig.4 –you don't have units on the y-axis or in the labels.*

Apologies – these will be added.

*Fig. 4 – Please define surface_r and interface_r. Is snow depth the difference between these two lines? Please clarify on the figure and in text.*

Yes, snow depth is the difference between the two lines and this will be stated.

*Fig. 5 –I think a diagram of an IMB would be helpful for modelers, which I think this one is, but it's poorly labeled.What do the dots represent (thermistors?) Is this why they go above the snow interface)? What does the L/R position of the dots mean (temperature?)? Why is there a green line over part of the dots?*

This figure is intended to illustrate the calculation of basal conductive flux and ocean heat flux from IMB data, as in Lei et al (2014), rather than being a more general diagram of an IMB. The green line therefore illustrates a linear fit through temperatures in the layer through which the basal conductive flux is calculated.

*Fig. 6 –the circle colors on this figure are very hard to make distinguish. Perhaps different shaped symbols in black would be better? Again, define surface_r, interface_r, and bottom_r. Are points where there is blue (aka surface_r) mean that the blue and green circles are overlapping? Adding arrows to indicate the transitions for the false bottom would also be helpful? This figure also makes me question why you don't linearly interpolate between the "correct" depth–it looks possible so why lose that data?*

We will carry out the corrections to the figure the reviewer suggests. Interpolating over the false bottom would probably work in this case. Such a step would need to be carried out at an earlier part of the analysis. However, interpolation might fail in other cases where

- false bottom formation was more gradual in onset
- the false bottom lasted for a longer time, concealing additional ice formation at the true ice base. In this case, analysis of the ice temperature might help, but it remains the case that false bottom formation greatly complicates the analysis.

*Fig. 7 – It might be clearer to show the total spread in values with shading and then just a central dot for the mean since the individual points with their spread get hard to distinguish. Also, in text you list the means but they need to be shown to be significantly different.*

We agree that this might make the figure clearer to read. But we would also then add an additional paragraph to discuss the uncertainty of the individual measurements, as these would not then be shown in any figure.

*Fig. 8 – What purpose does this figure provide other than the model bias, which we already expected from your previous work. Again, if you do show it, significance tests are important. I think a PDF of the flux by thickness would be more helpful to supplement this figure.*

It is true that the figure 'only' shows the model bias – but that is the whole point of a model evaluation, and the purpose of this paper is to demonstrate the value of this method of evaluation. Showing that the method is able to demonstrate model biases despite observational uncertainty, and that these biases are consistent with previous information, is vital to this.

We agree that significance tests would improve the evaluation, and that plotting conductive flux against thickness would be a useful exercise. As discussed above, it is difficult to obtain a true like-for-like comparison of mean thicknesses over ice in model grid cells to the point measurements of ice thickness to the IMBs. In the case of the conductive fluxes, the harmonic mean thickness over categories provides a meaningful comparison, but it is more difficult to see how these problems might be overcome when comparing the top melting and ocean heat fluxes by ice thickness.

*3) Code availability*

*The effort put in here by the authors to make the data available is lackluster. I understand the model code itself may not be available. However, the authors should make more effort to list how to get the buoy data (raw and processed) as well as the model data (if the code isn't available) since these should be public if it's part of the CMIP archive. See the guidance on the website:*
*https://www.geoscientific-model-development.net/for_authors/code_and_data_policy.html*

In fact the code can now be made available pending an internal review process, but this was not known at the time of initial publication.

The reviewer's suggestion that the processed IMB data should be made available was also made by Reviewer 1. As we agree that this would be a very useful step, we are currently overhauling the code in order to produce this data in a standard format (netCDF), and hope to publish this with the next revision of the paper, if one is requested.

***Minor concerns***

*Shu et al. 2015 isn't in your references.*

This will be added.

*Pg.4, line 6-8 – do you mean the ice temperature or air temperature? What's going on to cause the non-physical temperatures?*

Both can be affected, although the ice temperature is more likely to affect the ensuing analysis, and therefore more likely to be noticed and corrected. We assume that the cause is usually a sensor failure.

*Pg.4 line 14 -How long do buoys last? Give a range here.*

We will provide this.

*On page 7 you state the convention of positive fluxes indicate downwards. It would be very helpful if you mention that much earlier on Page 4 line 21. And put this on a diagram (Fig.1 and/or Fig.5) too.*

We will state this in the places you suggest (for the reasons above we think Figure 5 might be preferable to Figure 1 – this information may not be appropriate for an introductory schematic).

*Pg.7 line 20 –how thin is "quite thin"? Be specific!*

We will state the range of thicknesses for which this problem is relevant (very roughly, below 1.5m).

*Pg.7 line 26-28 –These means are over all available buoys and all years, right? It would be good to be explicit.*

Yes, they are over all buoys and all years.

*Pg.9 line 15 –change "7 to 143" W/m2 because negative fluxes are possible and the current wording is unclear.*

This will be changed.

*Pg.8 line 25 AND Pg.10 line 1, mention that those regions are defined in Fig.2.*

This will be mentioned.

*Pg.12, line 5 –what is the model grid(it hasn't been mentioned)?Why the huge range in grid cell size?*

The model grid is a regular latitude-longitude grid (the 'HadGOM grid'), with width one degree latitude and longitude throughout the world except in the tropics, where the latitudinal resolution is somewhat increased. Hence the range in grid widths in km in the Arctic is quite large, falling from ~40km near 70N to ~2km near the North Pole. The grid height, meanwhile, is one degree throughout (~110km).

**3. Summary of proposed changes and additions to manuscript arising from reviews**

The most substantial additional piece of work we will carry out is not directly linked to the manuscript. We will continue to overhaul the code in order to produce a processed dataset for publication, and the code will be published also after internal review.

**Major changes to the manuscript**

- The analysis will be extended to all four ensemble members of HadGEM2-ES, to allow model biases to be better set in context of internal variability.
- Uncertainty estimates on each monthly flux will be refined to allow for uncertainty in ice and snow density, and in fundamental constants of the relationship between temperature, salinity, conductivity and heat capacity. We will also expand the salinity ranges for both the top and lower surfaces of the ice.
- A discussion of the relationship between conductive flux and ice thickness will be added, probably with the aid of a scatter plot. For the reasons indicated a comparison of the relationships between ice thickness and top melt / ocean heat fluxes may be less useful, and will therefore probably not be included.

**Moderate changes**

- Where requested, the method descriptions in section 2 will be clarified, possibly with additional figures to demonstrate the process of top conductive flux estimation, as well as the more general IMB diagram requested by reviewer 2.
- The purpose of the paper will be clarified – the aim is not to investigate the model biases of HadGEM2-ES, but to demonstrate the value of a new method of model evaluation using HadGEM2-ES as a case study.
- The positive=downwards sign convention will be stated earlier and more frequently.
- We will attempt to process erroneous top melting and conductive fluxes prior to monthly averaging.

In addition, all minor changes indicated above will be made as suggested.

**References**

Pringle, D. J., Eicken, H., Trodahl, H. J., Backstrom, L. G. E.: Thermal conductivity of landfast Antarctic and Arctic sea ice, J. Geophys. Res. (Oceans), 112, C4, doi: 10.1029/2006JC003641

---

## Author Response (AR1)

**Authors' response associated with first resubmission of 'Using Arctic ice mass**
    **balance buoys for evaluation of modelled ice energy fluxes**

This response is in three parts:

a)  An updated point-by-point response to the reviewers. This is in the main very similar
  to the original response, but in some cases the actual changes made differ from the
  changes originally suggested.
b)  A summary of the changes to the manuscript.
c)  A tracked changes version of the manuscript.

**a)  Point-by-point response to reviewers**

Once again, we thank the reviewers for their effort in producing a helpful, constructive set of
suggestions for improving the manuscript. We apologise for the length of time it has taken to
produce a revision. This is mainly associated with an overhaul of the code (intended to allow
the data associated with the publication to be made more widely available), problems
encountered during this process, as described in the summary of changes below, and the
need for an internal review.

**1. Response to reviewer 1**

Original reviewer comments are shown in italic font, our response in normal font.

*General comments: The authors present a thorough analysis of the well known dataset of*
*ice mass buoys deployed covering the Central Arctic and Beaufort Gyre regions since 1993*
*to provide climatology seasonal estimates of the top and bottom ice conductive and melt and*
*ocean fluxes over and under sea ice. The novelty of the method lies in the fine analysis of*
*the data and in the physical processing applied to retrieve meaningful fluxes that can then be*
*used to evaluate climate models such as the HadGEM2-ES Met Office model. I am*
*supportive of this paper being accepted in this general as this dataset and methodology*
*offers a useful tool to the modelling but also remote sensing and in-situ communities.*
*Nevertheless one of the main strength of this paper is the (clean) dataset produced as well*
*as the algorithm developed to analyse these dataset and I strongly encourage the authors to*
*make these data available to the community. In addition,in this case, the nature of the*
*product calls for more transparency and sharing the code and data will warrant easier*
*reproducibility for the scientific community. Finally, a significant effort is needed to clarify the*
*sensitivity of the analysis to the variousconstants and approximations made. I provide some*
*detailed comments below on how this could be achieved.*

We thank the reviewer for their kind remarks. Although it was not possible to publish the
code in time for the publication of the discussion paper, due to internal procedures, we are
publishing a new version of the code with this revision of the paper.

As noted before, we strongly agreed with the reveiwer's suggestion that, in addition, a
version of the processed data should be made available to the community. Following this
suggestion, we overhauled the code to allow the data to be produced in netCDF4 format. In
addition, we divided the code into two stages, each with an associated output dataset. In the
first stage, IMB data is read, quality controlled, and mapped to a consistent set of data series at consistent time points. In the second stage, this processed data is used to calculate the datasets of monthly mean variables.

Because of this division into stages, there are two code repositories published with this paper (on GitHub) and two datasets (on Zenodo), one associated with each stage.

***Specific comments:***

***Abstract***

***Introduction***

*P1L23: add reference to Kwok, 2018*

This has been included in a rewritten Introduction.

***Calculating monthly-mean energy fluxes from the IMBs***

*P3L26: would a more advanced optimal interpolation scheme improve the results?*

Yes, probably – such a scheme could make use of information about other variables, for example. However we decided early on to use a very simple scheme for the present study to ease the processing of data. The results of the estimation scheme used were, individually, sensible. The estimation scheme is contained in a single function in the code, which could easily be replaced by a more advanced scheme in a future study. We have justified this choice with a sentence in our revision.

*P3L33: define z_srf and explain a little more (maybe in appendix or with figure 4) how zsrf and zint are sufficient to estimate both changes of surface and bottom sea ice.*

This point was not explained very clearly. z_sfc is now defined (surface elevation) and we have tried to improve the clarity of this paragraph.

*P4L2: King et al, 2018 and Mercouriadi et al, 2018 have shown that such snow ice formation is prevalent in some regions of the Arctic. Discuss.*

We have added a more detailed discussion of the snow-ice formation issue, noting the observations you mention (we have chosen to cite Rösel et al, 2018).

*P4L13: couldn't you ask the data providers?*

We asked the data providers, and now cite their response.

*P4L19: a link to the code would be very valuable here.*

The code is now linked.

*P4L26: you can cite Alexandrov et al, 2010 for values of snow and ice density. Snow evolves throughout the season with values typically from ~200 to 350 (i.e. Tilling et al, 2017)*

We have cited the paper the reviewer suggests. The new revision includes a more comprehensive uncertainty analysis, and as part of this we examine fluxes produced with snow densities of 274 – 374 kgm$^{-3}$, and ice densities of 917 – 944 kg m$^{-3}$; these seemed to us realistic boundary values given the conditions under which IMBs are deployed.

*P4L34: not clear where this formula comes from and if it applies to the real snow on sea ice. At what depth?*

As suggested in our initial response, we have rewritten this paragraph and provided a figure, and hope that the result is clearer.

*P5L9: this fixed thickness (say L) is a parameter of your analysis. Discuss how sensitive your results are to this choice.*

We have included this in our expanded uncertainty analysis.

*P5L29: similarly how does the uncertainty on these constants impact your results? Discuss.*

We have evaluated the impact of using a different scheme for estimating sea ice conductivity, that of Pringle et al (2006). As discussed above we also now evaluate uncertainty due to snow and ice density.

*P6L1: These values come from where? Recent work Nandan et al 2018 show salinity at the snow ice interface to be larger than 1. There are more references buy Turner et al 2015 (model work on CICE) but also Notz etc…*

We thank the reviewer for providing these references. Our view is that the subject of sea ice salinity is sufficiently complex that the only way of properly accounting for this in the present study, without seriously detracting from its main purpose, is to use uncertainty ranges that encompass all realistic salinity values. Hence in the revised paper, we use expanded uncertainty ranges in the next paper revision, allowing values up to 10 at both the top and lower surfaces of the ice.

*P6L5: equation is no readable*

This has been corrected.

*P6L17-21 where is that shown. Perform proper sensitivity analysis to all these parameters inyour plots, discussion etc…*

The reviewer has correctly identified that there are indeed additional sources of uncertainty not accounted for here, and we have tried to incorporate these into our expanded uncertainty evaluation section.

*P7L1: interesting. How would you inform the S value.Is it measured? Explain. What problem are you referring to here.*

Salinity is not measured, but the temperature and elevation data act to constrain the salinity ranges. For example, if the ice surface is at -0.1m, and the temperature at -0.2m is -0.1 deg C, this implies the melting temperature of the ice at -0.2m is greater than -0.1 deg C. Hence the salinity is lower than 1.9. The 'problem' as described in this paragraph is that occasionally the temperatures are in this way inconsistent with the assumed salinity ranges.

*P7L5: Tsamados et al, 2015 has implemented the three equation boundary conditions and discussed false bottom impact on sea ice - ocean bottom fluxes*

Instead of our stating that false bottom formation renders the computation of ocean heat flux impossible, it is probably more accurate to say that it greatly complicates its calculation – and given the number of data points affected, may be outside the scope of this study. We decided not to add a reference to Tsamados et al because it seemed to us that this study concerned the indirect impact of false bottoms on the ice energy budget, rather than the direct impact.

*P7L13: Interesting. Can you see synoptic signal related to snow forcing (i.e. storms?). At what timescale are you solving these? Monthly? Should you pre-process such erroneous signals before monthly averaging? Explain -> share code!*

In the overhauled code, we have switched to removing 'cold' top melt data prior to monthly averaging.

**Deriving monthly-mean flux distributions from the IMBs**

*P7L27: why not two regions in the table*

We did not want to include too much information in a single table, to improve ease of reading. In the revision, we have provided two additional tables, giving the fluxes by region.

*P7L28: why don't you discuss changes between decades I.e. 90s vs 00s vs 10s?*

We decided that analysis of interannual variability in the IMB fluxes would be a valuable addition to the paper; this is in a new subsection (3.2). However, this is difficult to perform satisfactorily because of a lack of data points. Only 7 buoys are available for the 1990s; 6 of these were from the SHEBA campaign (i.e. in the same year, 1997-1998, at the same location in the Beaufort Sea), and the remaining buoy, deployed in 1993, was also located in the Beaufort Sea. Hence there is not enough data from this decade to properly sample spatial or interannual variability in the Beaufort Sea region, and none at all in the North Pole region. We chose instead to use a series of irregular periods to compare, with roughly comparable numbers of data points. However, there is still not enough data, or variation, to unequivocally detect interannual variability, and we state this.

*P9L1: explain a bit more how these errors on the individual monthly scatter points are obtained. Are you performing an error propagation or are these simply a standard deviation?*

This point is superseded by the new uncertainty analysis in subsection 3.3. We do not use scatter points any more in the figure in question, instead demonstrating the IMB dataset with a series of boxplots.

**4 Evaluating modelled sea ice using the. IMB-derived fluxes**

*P9L21: not clear if you estimate the fluxes at the same location in time and space as the IMBs or average over the whole region for the whole month. You should both to test impact of IMB sampling on your results.*

We average over the whole region. Model internal variability is such that we do not expect the model to exactly capture the conditions at each point in space and time, and therefore did not see any particular value in sampling the model only at identical points to the IMBs. As discussed in the appendix, we suspect that the largest impact of IMB sampling is through the ice thickness – this would not be solved by evaluating the model at the same points in space and time, as we would still be evaluating fluxes over the entire grid cell. It was not clear to us, therefore, that the impact of IMB sampling would be revealed by the methods that the reviewer suggests. We have included a paragraph to this effect in our revision.

*P9L27: why didn't you perform your analysis on a more advanced model with more Ice thickness categories?*

We chose to evaluate HadGEM2-ES because its sea ice simulation was already fairly well-understood. Confidence in the IMB-based evaluation could therefore be informed by how consistent this was with the sea ice and surface radiation evaluation. We are now evaluating the new UK CMIP6 models (HadGEM3-GC3.1 and UKESM1.0) in the same way, but this evaluation appears to be outside the scope of this study, which is intended only to demonstrate the new method. Note that although the new models are more advanced (multilayer thermodynamics and explicit meltponds), the number of thickness categories is the same.

*P9L35: is it West2018 or 2019.*

2019. This has been corrected.

*P10L4: again not clear if you perform comparison like for line (i.e. forsame days and grid cells) or not.*

No – the model distribution is calculated over the whole region, for reasons described above. Model internal variability means that we do not expect fluxes at the exact same pathways, at the same times, to better represent the conditions than the fluxes over the whole region and time period.

*P10: here you list various fluxes but don't explain why you find these results. A bit too descriptive.*

We have added some discussion of the results in this section.

*P11L5: West 2018 or 2019?*

– again this hs been corrected.

*P11: discuss role of melt ponds (summer) and snow cover (winter)*

The role of snow cover in winter in the conductive flux biases can be investigated directly by comparing IMB-measured snow depths to those modelled by HadGEM2-ES. We have carried out such a comparison alongside a comparison of conductive flux and ice thickness which was requested by Reviewer 2 (see below), in a new subsection 4.3.

The role of melt ponds in summer in the top melt biases (which is what we assume the reviewer means) is more difficult to evaluate directly using the IMBs. However, the meltpond parameterization of HadGEM2-ES was strongly implicated in causing the net surface flux biases found by West et al (2019), and is hence probably implicated in the top melt biases found by this study. We have noted this.

*P12: I think a lot of your analysis is missing the link to melt pond coverage*

We apologise for omitting to respond to this point in our earlier response.. We have discussed the role of melt ponds in the HadGEM2-ES top melt bias, as indicated above. If Reviewer 1 has additional suggestions to improving this aspect of our analysis we will be very happy to consider how to implement them.

**2. Response to reviewer 2**

*The authors use ice mass balance buoys (IMB) to estimate fluxes through the top and bottom of sea ice. The authors present this new method and then compare the observed fluxes in the North Pole and Beaufort Sea regions. The authors then compare the observed fluxes with modeled fluxes from the HadGEM2-ES climate model. The main findings are that there are biases in these fluxes in the model, which are likely due to the biased mean state of the model. Additionally, there are differences in the fluxes in the Beaufort Sea and North Pole regions. I havea fewm ajorand moderate concerns about the way the model and observations are compared and these need to be addressed before I can recommend publication.*

*Major concerns*

*1) Internal Climate Variability*
*The elephant in the room for a comparison between a climate model and observational data is the issue of internal climate variability (see references below), which you never mention, and leads me to have major concerns with your method. This is also relevant for when the Arctic will become ice free, which you mention in the introduction. Since HadGEM2-ES is a fully-coupled, freely-evolving climate model a single model experiment should not be expected to match the observed sea ice conditions. You do not mention using ensembles and where/how the observations fit in an ensemble spread.*

We have expanded the analysis to include the full ensemble (albeit only 4 members) to estimate the internal variability in the fluxes, and compare these to the model biases. The differences are described below, in the summary of changes (briefly, no changes to the model means, but some difference to standard deviation in the top melt flux).

As before we note that the main purpose of the study is to demonstrate the value of the IMB-based evaluation, rather than to draw conclusions from the model biases demonstrated. To demonstrate the value of the evaluation method, it is in our view sufficient that a) the IMB-measured model biases are consistent with biases in the sea ice and surface radiation simulations, and b) they are larger in many months than the observational uncertainty in the IMB fluxes. Whether or not the biases result from internal variability, or another cause, is in our view of only secondary importance for this study. This would not be the case if we were trying to draw conclusions from the model biases, in which case the internal variability context would be vital (hence the many studies of internal variability in sea ice extent trends, some of which the reviewer quotes).

We have rewritten the abstract and introduction to try and clarify this purpose.

*Indeed, it appears you are comparing the mean climate model state with the mean observations (Fig. 8). This is not a particularly useful comparison –we know the model is biased from your previous work and therefore we expect to see biases in these mean fluxes as a result!*

But we would argue that to find biases in the mean fluxes that are physically consistent with biases in the sea ice simulation is in itself a significant result – because we are demonstrating an entirely new method of model evaluation. We have stated this more explicitly.

*Instead, a more useful analysis would be to evaluate situations when the model does have similar thicknesses to those observed, do the fluxes match the observations? That would tell us more about the processes going on in the model and how well they compare to those*

*observed. You could do this by plotting the distribution of the conductive fluxes by thickness for the model and observations for a particular month or throughout the year.*

A comparison of conductive fluxes to ice thickness, for both the model and IMBs, has been added in a new subsection 4.3. In addition we compare conductive fluxes to snow depth, as suggested by Reviewer 1.

*Two additional comments on the model: a) It would be useful to quantify other relevant model biases to the sea ice mass budget like SST or ocean heat transport.*

Ocean heat transport has been quantified in our revision, in a paragraph following the evaluation of modelled ocean heat flux. We quantify OHC over the Arctic Ocean as a whole, and also over the North Pole and Beaufort Sea regions, and discuss the likely nature of the link between this and ocean-to-ice heat flux.

*b) You compare different years from the model (1980-1999) and observations (1997-2016). I know you did analysis about how the periods are different (Pg.12, lines 13-24) but why not just use the same years that presumably have comparable radiative forcing?*

We chose the period of 1980-1999 for consistency with the earlier study of HadGEM2-ES sea ice and surface radiation. The historical ensembles of HadGEM2-ES actually end in 2005, so it would be necessary to use a scenario experiment to get a comparable time period.

*2) Additional sources of uncertainty*

*You mention uncertainty in salinity as one of the big uncertainties in the IMB flux calculations. I think you need to mention that there are also large ranges in the observed snow and ice densities (see refs below) that could cause uncertainty in the retrievals. The values you use are reasonable, but you need to at least acknowledge this and do some basic calculations about how big a difference these values make. Franz et al. 2019 (doi: 10.5194/tc-13-775-2019) Webster et al. 2018 (doi: 10.1038/s41558-018-0286-7)*

We have included an analysis of uncertainty due to ice and snow densities in our revision, as part of an expanded uncertainty analysis (new subsection 3.3). We chose ranges of 274-374 $kgm^{-3}$ (snow) and 917-944 $kgm^{-3}$ (ice).

*3) Significance*

*You spend much of section 3 describing differences in observations in two regions and sections 4.1/4.2 describing differences in the mean state of the model and observations. No significance tests were discussed to indicate whether themeans are really significantly different in Figs. 7/8. A simple t-test should suffice, but without this I have a hard time believing some of the conclusions (e.g. that September fluxes differ in the IMB between regions).*

This appeared an excellent suggestion, and significance tests have been incorporated throughout sections 3 and 4. A Welch t-test appeared to us to be appropriate, as in most cases the sample sizes will not be the same and the variances of the distributions cannot be assumed to be equal. We have

**Moderate concerns**

*1) Model thermodynamics*

*I am surprised that HadGEM2-ES, a CMIP class climate model, uses the very simple zero-layer thermodynamics and I think that this should be addressed. The Bitz and Lipscomb thermodynamics is more realistic than zero-layer, and even this has been superseded by the mushy layer thermodynamics of Turner and Hunke (see below). I realize you can't change the model at this point and this shouldn't prevent publication, but your own results show that the assumptions of the zero-layer scheme for conductive fluxes are bad (Fig. 7 and Table 1). Maybe one of the conclusions should be that the zero-layer cannot represent the observed processes so HadGEM2-ES might want to stop using it?*

*Bitz and Lipscomb 1999 (doi: 10.1029/1999JC900100) Turner and Hunke 2015 (doi: 10.1002/2014JC010358)*

The new UK CMIP6 models, HadGEM3-GC3.1 and UKESM1.0, use the multilayer thermodynamics formulation of Bitz and Lipscomb (1999); a further paper is planned to compare these to HadGEM2-ES. But our view was that they are not relevant for the present study, which is intended to demonstrate a new method of evaluation rather than to compare models.

*2) Figures*

*Individually these comments are fairly minor, but the sum of them is moderate.*

*• Table 1 –Please add the # values per month and per flux to this table rather than just listing them in the text. Also, adding the units below the flux names (not just in the table description) would be helpful. I expect top melt to usually be in cm/day (or kg m-2s-1), not W/m2.*

Adding number of samples and units are both good suggestions to aid clarity, and have been carried out. For the reasons given in our original response however we have left the units unchanged.

*Fig. 1 – It would be helpful to add the sign convention for fluxes here at the interfaces (show the flux direction for positive!). Either label or remove the red/yellow arrows.*

The red and yellow arrows have been removed as it was considered they did not add useful information for a diagram that was purely for illustration of processes involved. We were not sure that a sign convention was necessary here for similar reasons.

*Fig.3 and Fig.4 –you don't have units on the y-axis or in the labels.*

Apologies – these have been added (now figures 4 and 5).

*Fig. 4 – Please define surface_r and interface_r. Is snow depth the difference between these two lines? Please clarify on the figure and in text.*

We have improved the labelling on this figure (now figure 5).

*Fig. 5 –I think a diagram of an IMB would be helpful for modelers, which I think this one is, but it's poorly labeled.What do the dots represent (thermistors?) Is this why they go above the snow interface)? What does the L/R position of the dots mean (temperature?)? Why is there a green line over part of the dots?*

We have added a diagram of an IMB (figure 2) and incorporated most of the information on the original figure 5 onto here.

*Fig. 6 –the circle colors on this figure are very hard to make distinguish. Perhaps different shaped symbols in black would be better? Again, define surface_r, interface_r, and bottom_r. Are points where there is blue (aka surface_r) mean that the blue and green circles are overlapping? Adding arrows to indicate the transitions for the false bottom would also be helpful? This figure also makes me question why you don't linearly interpolate between the "correct" depth–it looks possible so why lose that data?*

We have carried out the corrections to the figure the reviewer suggests, but the figure has been moved to supplementary material to make room for other figures.

Interpolating over the false bottom would probably work in this case. Such a step would need to be carried out at an earlier part of the analysis. However, interpolation might fail in other cases where
   • false bottom formation was more gradual in onset
   • the false bottom lasted for a longer time, concealing additional ice formation at the true ice base. In this case, analysis of the ice temperature might help, but it remains the case that false bottom formation greatly complicates the analysis.

*Fig. 7 – It might be clearer to show the total spread in values with shading and then just a central dot for the mean since the individual points with their spread get hard to distinguish. Also, in text you list the means but they need to be shown to be significantly different.*

We have replaced the scatter points with boxplots, and have added a subsection in which uncertainty due to various parameters is evaluated in detail.

*Fig. 8 – What purpose does this figure provide other than the model bias, which we already expected from your previous work. Again, if you do show it, significance tests are important.I think a PDF of the flux by thickness would be more helpful to supplement this figure.*

We agreed that significance tests, and an examination of the relationship between conductive flux and ice thickness, would both be valuable enhancements of the study, and have added these.

It is true that the figure 'only' shows the model bias – but that is the whole point of a model evaluation, and the purpose of this paper is to demonstrate the value of this method of evaluation. Showing that the method is able to demonstrate model biases despite observational uncertainty, and that these biases are consistent with previous information, is vital to this.

*3) Code availability*

*The effort put in here by the authors to make the data available is lackluster. I understand the model code itself may not be available. However, the authors should make more effort to list how to get the buoy data (raw and processed) as well as the model data (if the code isn't available) since these should be public if it's part of the CMIP archive. See the guidance on the website: https://www.geoscientific-model-development.net/for_authors/code_and_data_policy.html*

The code, and associated data, is published alongside this revision. We have overhauled the code to enable production of data in netCDF4, and have divided the code into two stages, each with associated output data. In the first stage, IMB data is read, quality controlled, and processed into a consistent set of data series on consistent time points. In the second stage, monthly mean energy fluxes are calculated from the resulting data. Each stage has an associated code repository on GitHub and an associated dataset repository on Zenodo.

***Minor concerns***

*Shu et al. 2015 isn't in your references.*

This has been added.

*Pg.4, line 6-8 – do you mean the ice temperature or air temperature? What's going on to cause the non-physical temperatures?*

We contacted the data providers to confirm this, and have added their response.

*Pg.4 line 14 -How long do buoys last? Give a range here.*

We have provided this.

*On page 7 you state the convention of positive fluxes indicate downwards. It would be very helpful if you mention that much earlier on Page 4 line 21. And put this on a diagram (Fig.1 and/or Fig.5) too.*

We have stated this in the places you suggest (for the reasons above we thought Figure 5 might be preferable to Figure 1 – this information may not be appropriate for an introductory schematic).

*Pg.7 line 20 –how thin is "quite thin"? Be specific!*

In fact we have removed this paragraph, and associated quality control check, as on reflection we did not think it was important.

*Pg.7 line 26-28 –These means are over all available buoys and all years, right? It would be good to be explicit.*

Yes, they are over all buoys and all years.

*Pg.9 line 15 –change "7 to 143" W/m2 because negative fluxes are possible and the current wording is unclear.*

This has been changed.

*Pg.8 line 25 AND Pg.10 line 1, mention that those regions are defined in Fig.2.*

This has been mentioned.

*Pg.12, line 5 –what is the model grid(it hasn't been mentioned)?Why the huge range in grid cell size?*

The model grid is a regular latitude-longitude grid (the 'HadGOM grid'), with width one degree latitude and longitude throughout the world except in the tropics, where the latitudinal resolution is somewhat increased. Hence the range in grid widths in km in the Arctic is quite large, falling from ~40km near 70N to ~2km near the North Pole. The grid height, meanwhile, is one degree throughout (~110km). This information has been provided in the model description.

                 **Summary of changes to manuscript**

**1.  Brief summary of changes**

The most substantial change to the manuscript arises from the overhaul of the code used to
produce the IMB fluxes, to enable production of output data in netCDF format (the code itself
is also now published). In the course of this, some bugs and problems were identified and
corrected that have resulted in changes to the IMB dataset, that are in the main small. There
are also three substantial new areas of analysis: interannual variability in the IMB fluxes,
more comprehensive evaluation of uncertainty in the IMB fluxes, and an investigation of the
relationship between conductive fluxes, ice thickness and snow depth. In addition, the
analysis of internal model variability is improved by including all 4 ensemble members of
HadGEM2-ES.

These five major modifications are described in more detail in subsection 2 below. In
subsection 3, a complete line-by-line description of all changes to the manuscript is
presented.

**2.  Description of major changes to the study**
**2.1 Overhauled code, and resulting changes to the IMB dataset**

Strong desires were expressed by both reviewers that code and data should be published
with any revision of the study. While permission was obtained to publish the code soon after
the initial submission, no consideration had previously been made of publishing the
associated data, which had not been produced in a particularly commonly-used format.
Hence it was decided to overhaul the code to enable production of the IMB data in netCDF4
format. In the course of this, the code was divided into two repositories, each representing
different stages of the analysis. In the first stage, the IMB data is read, quality controlled,
interpolated or averaged to temperature measurement points, and processed to a standard
set of data series, before being written to netCDF4. In the second stage, the processed data
is used to produce a set of monthly mean energy fluxes, using the processes described in
section 2. The code of each stage is published in separate GitHub repositories, and the two
resulting netCDF4 datasets are published in separate linked Zenodo repositories.

There are a number of small differences in the new IMB dataset relative to the old, resulting
from a mixture of bugfixes and necessary changes. Specifically:

- All fluxes in the new dataset are produced from timeseries regularised to temperature
measurement points. Before, some series were produced from timeseries regularised
to daily time points (usually midnight GMT). In the case of the conductive fluxes, this
produced a bias when a diurnal cycle was present in the temperature data. Use of
the temperature measurement points, which tend to be bi-hourly, removes this bias.
This causes large differences in particular in the top conductive flux, especially during
the spring.
- The use of different time points also causes missing data points to be treated slightly
differently, such that months which may have reported valid data in the old dataset no longer do so (because temperature was not measured correctly for a significant portion of that month).

- It was found that salinity had not been properly taken into account when calculating latent heat of freezing of the basal surface of the ice; this has been remedied in the new dataset, and generally results in slightly higher ocean-to-ice heat fluxes.
- At the base of the ice, the salinity uncertainty range now takes ice temperature into account. While over most of the dataset, a salinity range of 0-10 is used, salinities which imply a melting temperature below the actual temperature of the ice are ruled out. This avoids the occurrence of singularities which previously had to be manually identified after processing and removed from the dataset.
- As suggested by Reviewer 1, removal of erroneous top melting data is now performed before monthly averaging, instead of afterwards. Instances of falling surface elevation are now judged due to top melting only when surface temperature exceeds a fixed threshold (-2°C), being otherwise assigned to a separate data series.

The cumulative effect of these changes does not change the qualitative properties of the IMB dataset discussed in Section 3, but it does lead to small changes in the numbers shown in Table 1. There is also no effect on the qualitative statements about model biases in Section 4, with the exception of the (small) ocean heat flux bias in the Beaufort Sea region which is eliminated by the change to latent heat calculation. This bias was only incidental to the paper's conclusions as it was small compared to the biases in top melting and conductive fluxes.

**2.2 Interannual variability**

As indicated in the original response to the reviews, there are not enough data points to permit a year-by-year comparison of the IMB data. Instead, the dataset was divided into three periods of roughly equal data points and distributions from each period compared. To accommodate this analysis, section 3 was divided into subsections. The previous evaluation of seasonal and spatial variability in the IMB fluxes is now in section 3.1, and the interannual variability analysis is in a new section 3.2.

**2.3 Improved analysis of uncertainty**

In the original version of the paper, we analysed uncertainty in the IMB fluxes due only to salinity, where salinity was assumed to be 1 ± 1 near the upper surface of the ice, and 4±4 at the base. The reviewers made a number of suggestions as to how this could be improved. Based on these suggestions, in the revision we have also examined the impacts of

- Using another widely-used scheme to estimate ice conductivity, based on Pringle (2007)
- Using different reference layers to calculate conductive fluxes and ocean heat flux
- Using different values for ice and snow density, based on Alexandrov et al (2010) and its sources Cox and Weeks (1982) and Romanov (1995)
- Using larger ranges for ice salinity (from 0 – 10).

The evaluation performed in sections 3.1 and 3.2 is that for the 'standard configuration': Maykut and Untersteiner conductivity scheme, a reference layer of .4m-.7m above the ice base to calculate conductive fluxes, and the original values for snow and ice density (330 and 917 kgm-3), but with error bars shown over all possible values of salinity (as before). In a new subsection, 3.3, we then explore the impact on these results of altering in turn the
parameters above.

**2.4 The relationship between conductive flux, sea ice thickness and snow depth**

Both reviewers suggested that an analysis of the link between conductive flux and sea ice
thickness would be of value, to judge whether the conductive flux biases of HadGEM2-ES
were entirely caused by ice thickness biases. In addition, Reviewer 1 requested an analysis
of the role of snow depth. Both are examined in a new subsection 4.3. IMB and model data
points are now sorted into distributions based on intervals of (in turn) ice thickness and snow
depth, and variation in conductive flux distributions with these variables is examined.

**2.5 Internal variability**

To more fully assess the impact of internal variability on the results (as requested by
Reviewer 2), we evaluated the full HadGEM2-ES historical ensemble (4 members); in the
original manuscript, only the first member was evaluated. The addition of the three other
members did not significantly alter mean fluxes in any instance, (so the model biases were
qualitatively unchanged). However, standard deviation of top melting flux did rise
significantly in July and August in the North Pole region, with standard deviation twice as
high in July (44.0 compared to 20.1 Wm-2) and nearly five times as high in August (53.1
compared to 11.2). No comparable effect is visible in any other variable and region, and
given the positive-definite nature of the top melt flux it is likely that this effect is produced by
a handful of model years (confined to ensemble members 2-4) with exceptionally high top
melting in the North Pole region, thus producing a highly skewed overall distribution of
fluxes.

| -------------------- | First ensemble member only | | Whole ensemble | |
|---|---|---|---|---|
| Topmelt (Wm$^{-2}$) | Mean | Std. | Mean | Std. |
| 1 | 0.0 | 0.0 | 0.0 | 0.2 |
| 2 | 0.0 | 0.1 | 0.0 | 0.1 |
| 3 | 0.0 | 0.0 | 0.0 | 0.1 |
| 4 | 0.1 | 0.3 | 0.1 | 0.4 |
| 5 | 3.0 | 3.5 | 3.5 | 3.7 |
| 6 | 56.6 | 14 | 55.8 | 14.3 |
| 7 | 57.2 | 20.1 | 56.6 | 44.0 |
| 8 | 11.7 | 11.2 | 11.9 | 53.1 |
| 9 | 0.3 | 5.5 | 0.4 | 4.0 |
| 10 | 0.1 | 0.4 | 0.1 | 0.4 |
| 11 | 0.0 | 0.0 | 0.0 | 0.8 |
| 12 | 0.0 | 0.1 | 0.0 | 0.2 |

**3. Complete description of changes to manuscript**

In the following description, line and page numbers refer to the non-tracked changes version.

**Abstract**

P1, L8: The abstract has been rewritten, to make the purpose of the study clearer: to demonstrate a new method of sea ice model evaluation.

**Introduction**

P1, L25: The Introduction has been rewritten for similar reasons, with the citation suggested by Reviewer 1 (Kwok, 2018) added at P1, L33.

**Calculating monthly mean energy fluxes from the IMBs**

P3, L5: A new figure, with a diagram of the main components of an IMB, is referenced (Reviewer 2 suggestion).

P3, L12: The range of buoy lifetimes is described.

P3, L25: Justification of interpolation scheme added.

P3, L30: This paragraph (description of the process of estimating interface from surface) has been reworded for clarity, and zsfc is now defined in the text.

P4, L5: Reasoning for rarity of snow-ice formation in the IMB dataset is discussed in more detail.

P4, L15: Possible reasons for failure of temperature sensors are discussed, following communication with the data providers, as suggested by Reviewer 1.

P4, L20: We note here the sign convention, as suggested by Reviewer 2.

P4, L20: The paragraph discussing assumptions about the elevation of the top temperature measurement point has been deleted, as all such elevations are now known following communication with the data providers.

P4, L24: Incidence of surface elevation decrease accompanied by cold temperatures is now removed from the top melt dataset prior to monthly averaging, and that is made clear here.

P4, L26: The values of ice and snow density used have been moved from here to the thermodynamic parameters section, for consistency.

P4, L29: The top conductive flux section has been rewritten for clarity, with the additional of a new figure (Figure 6).

P5, L18: When the reference layer used for the basal conductive flux is defined, it is noted that we examine the sensitivity to this reference layer elsewhere in the study.

P5, L25: a reference to the new Figure 2 (IMB diagram) is added.

P6, L3: The new approach to evaluating uncertainty is outlined: the use of a standard set of parameters to calculate the main dataset, and calculation of the dataset with altered values of parameters.

P7, L5: Sources of uncertainty examined later in the study are summarised.

P7, L14: This paragraph (temperatures too warm for the assumed salinity) has been reworded for clarity. In addition, a slightly different approach is used in the revised code: when the data used to calculate a monthly flux contains temperatures too high for the assumed salinity, the highest salinity consistent with the observed temperatures is used instead.

P7, L21: figure reference changed to supplementary material

P7, L26: Sentence added discussing the possibility of interpolating over false bottoms.

**Description of monthly mean fluxes from the IMBs**

P7, L31: Title changed for greater contrast with the title of Section 2.

P7, L32: This section has been split into subsections, to allow for separate analysis of interannual variability, and a more comprehensive analysis of uncertainty due to parameters of the analysis.

P7, L36: In the dataset description section, there are many changes to the numbers (mostly small), related to the code changes discussed above. These are not individually listed here.

P8, L3: A reference to the strong positive skew in the top melting dataset is added.

P8, L33: New tables have been added (Tables 2 and 3) to describe the dataset in the North Pole and Beaufort Sea regions separately (Reviewer 1 suggestion).

P8, L33: The distributions are now visualised using boxplots, rather than a large number of individual points.

P8, L35: Throughout this subsection, references to the significance of differences between distributions, evaluated using Welch t-tests, are added.

P9. L36: New subsection added discussing interannual variability in the dataset.

P10, L33: New subsection added, describing the sensitivity of the derived monthly fluxes to parameters of the analysis: salinity, snow density, ice density, conductivity and reference layer chosen.

**Evaluating modelled sea ice using the IMB-derived fluxes**

P13, L5: Reference to the grid used by HadGEM2-ES added.

P13, L8: West et al (2018) changed to (2019), here and below.

P13, L17: Reference to significance test now used to compare modelled and observed distributions added.

P13, L21: The approach of using model cells from the entire region in the comparison is justified.

P13, L29: Discussion of significance of differences is added throughout this section.

P13, L37: Here, and in many other places, more discussion of the results is added, in recognition of Reviewer 1's point that it was previously too descriptive. In particular, the likely role of the HadGEM2-ES meltpond scheme in the top melting biases is discussed, following the suggestion by Reviewer 1 that the role of meltponds should be examined in more detail.

P14, L34: After the evaluation of ocean heat flux, a paragraph examining the role of oceanic heat convergence is added (Reviewer 2 suggestion).

P15, L15: The model biases are compared to the dataset uncertainties evaluated in section 3.3.

P15, L33: Additional reference to the role of the HadGEM2-ES meltpond scheme inserted.

P16, L19: New subsection added examining the relationship between conductive flux, ice thickness and snow depth in model and IMB observations.

**Representativity of the IMB fluxes**

**Conclusions**

P18, L25: This section has been rewritten to include additional conclusions resulting from

    a) Interannual variability analysis
    b) More comprehensive evaluation of uncertainty in the IMB dataset
    c) The comparison of conductive flux, ice thickness and snow depth

**Appendix A**

**Code availability**

P22, L29: References to the two GitHub repositories where the IMB analysis code is now stored have been added. However, code used in producing figures and tables for the paper is still provided as a .tar supplement (except Figures 1 and 2 which were produced in Powerpoint).

**Data availability**

P22, L37: References to the Zenodo repositories where the processed IMB data is now stored have been added. A clarification is made regarding the model data: the code can be made available upon request to the author.

**Tables**

Table 1 has been updated to include number of observations of each flux, as suggested by Reviewer 2. As noted above, most values have changed slightly due to the issues discussed in the code changes section.

Tables 2 and 3, detailing fluxes in the North Pole and Beaufort Sea region, are new in this revision.

**Figures**

Arrows have been removed from Figure 1, in line with the suggestion of Reviewer 2; we agree that these added no useful information.

Figure 2, a diagram of an IMB, has been added (Reviewer 1 suggestion). The information included in the previous Figure 5, indicating reference layers, has been added to this figure instead and is referenced from later in the study.

Figure 4 (previously Figure 3): units added

Figure 5 (previously Figure 4): units added, and line labelling improved.

A new figure 6 has been added, demonstrating the process of calculating top conductive flux in the event of the reference layer lying partially within snow and partially within ice.

The previous Figure 6 has been moved to supplementary material to help allow for 3 new figures. The edits suggested by Reviewer 2 have been implemented.

Figure 7: instead of plotting individual points, we are now showing boxplots.

Figure 8 is new for this revision, and shows interannual variability in the IMB fluxes (Section 3.2).

Figure 9 (previously Figure 8): For consistency with Figure 7, and because of high skew in some of the fluxes, boxplots are now used instead of mean and standard deviation bars.

Figure 10 is new for this revision, and shows the relationship between conductive flux, ice thickness and snow depth in IMBs and model.

The previous Figure A1 has also been moved to supplementary material.

**c) Tracked changes version of manuscript**

[revised manuscript text omitted]

---

## Author Response (AR2)

**Authors' response to editor comments on 'Using Arctic ice mass balance buoys for evaluation of modelled ice energy fluxes'**

We thank the editor for their additional helpful suggestions for our manuscript. A point-by-point response follows, with a description of changes made. A tracked changes version of the manuscript is appended below.

*Comments to the Author:*

*\* Could the versions of the code, as used in the paper, be archived (e.g. in a Zenodo repository) as per the archiving information at https://www.geoscientific-model-development.net/about/code_and_data_policy.html.*

We have archived the figure code at https://zenodo.org/record/3947782#.XxAi9XdFyUk, and linked this from the 'Code availability' section.

*\* Figure 2 caption: Perhaps move the reference to the bibliography? Also I note the data citation details given at http://imb-crrel-dartmouth.org/results/.*

We have found a publication with a similar figure (Planck et al, 2019), with help from the data providers, and referenced this instead in the Figure 2 caption. We have added the data citation to the bibliography and referenced this at the beginning of Section 2 and in the 'Data availability' section.

*\* Page 3 line 11 and data availability section: I was not able to access this link -- should this be updated?*

Yes, it should have been http://imb-crrel-dartmouth.org/results/. We have updated this in both locations.

*\* Figure 6: Can you clarify \alpha k_{ice/snow} here?*

We have clarified the symbols here in a sentence at the end of the figure caption.

*\* Section 3 title "meanflux"*

This has been corrected.

*\* Page 9 line 25 and elsewhere, unit typesetting*

We have corrected multiple instances of this mistake, including in the Figure 10 labelling.

*\* Page 12 equation on line 6, units?*

We have clarified the units for T and S here.

*\* Figures S1 and S2 appear to be missing*

These should have been in the supplement; we are not sure what went wrong here. As the figure code is no longer needed, we have provided a new supplement which contains only Figures S1 and S2, and hope that these appear as required.

In addition to the above changes, we have thanked the anonymous reviewers in the 'Acknowledgements' section, and numbered the equations in the main body of the paper.

A tracked changes version of the manuscript follows:

[revised manuscript text omitted]

---

## Author Response (AR3)

**Authors' response to editor's suggested corrections to 'Using Arctic ice mass balance buoys for evaluation of modelled ice energy fluxes'**

A point-by-point response to the suggested corrections follows, and the tracked changes version of the manuscript begins on the following page.

*I think the units of k_I are missing for equation (6).*

The units (Wm$^{-1}$°K$^{-1}$) have been added to the definition of k_I in the preceding line.

*Perhaps clarify "Figure S1 (supplement)" and "Figure S2 (supplement)"?*

These have been clarified (the S1 reference using comma rather than brackets as the reference itself was already in brackets).

In addition, a new contribution has been noted in the 'Author contribution' section.

[revised manuscript text omitted]